

# On the relationship between static stability and anvil clouds

Zhenquan Wang[1, *]

[1] School of Atmospheric Sciences, Nanjing University, Nanjing, China

*Correspondence to:* Zhenquan Wang (zhqwang@smail.nju.edu.cn)

**Abstract.** The climate feedback mechanism of the anvil amount is still uncertain but is a key component in understanding the Earth's climate sensitivity. The environmental cloud-controlling factors are helpful to reveal the underlying processes of the anvil climate responses and constrain the climate model projections. The 3-km-layer stability below the lapse-rate tropopause is defined as the upper-tropospheric stability (UTS) and usually used to analyze the environmental stability control on the anvil clouds. However, the UTS might underestimate the stability control on the anvil. Here, a novel estimated anvil-top stability (EAS) based on the minimum stability in the upper troposphere is proposed as a stronger anvil-controlling factor than the UTS. By the radiosondes and cloud observations at the tropical Manus ground-based site, the results show that the EAS can better stratify the ice cloud incidence from 0.16 to 0.91 with a negative linear relationship, while the UTS only controls the ice cloud incidence from 0.4 to 0.78 and their relationship is nonlinear. The EAS can also be derived from the pressure-level reanalysis data, but due to the relatively coarse vertical resolution of the reanalysis the EAS would not be perfectly reproduced from the reanalysis. In comparison, the reanalysis-based EAS has a correlation of 0.51 with the radiosonde-measured EAS but it is still a stronger anvil-controlling factor than both the reanalysis-based and radiosonde-measured UTS.

At the global scale, the relationships of the anvil clouds with the reanalysis-based EAS and UTS are examined by the satellite passive and active sensors, respectively. As observed by the passive radiometer imagers on board the geostationary satellites, the distribution of the high cloud coverage (HCC) above 300 hPa has the correlation of -0.6 with the EAS but the correlation with the UTS is only -0.01. Daily HCC variations are strongly correlated with EAS with the mean correlation of -0.52 in the HCC domains, while the UTS underestimates the stability controls on the HCC (the mean correlation is -0.33). By the combined active sensors of the radar and lidar on board the polar-orbit satellites, the results show that the ice cloud fraction profiles in tropics, subtropics and midlatitude are all linearly stratified by the EAS, while the large ice cloud fraction could happen in both the small and large UTS and the relationship of the ice cloud fraction with the UTS is nonlinear. With the strong linear correlation to the anvil, the EAS can be used as a good predictor to understand the anvil climate feedback processes.

## 1. Introduction

Cloud responses to the environmental changes have not been correctly simulated in models (Gettelman and Sherwood, 2016; Webb et al., 2012; Sherwood et al., 2020; Wall et al., 2017), due to the imperfect model parameterization for the highly interactive processes of radiation, convection and cloud physics (Powell et al., 2012; Zhao, 2014; Bretherton, 2015; Zhao et al., 2016; Atlas et al., 2024; Suzuki et al., 2013; Klein et al., 2017). To reduce the uncertainty about the cloud climate feedbacks, the climate model predictions of the environmental changes are combined with the emergent constraints of the observed linear relationship between clouds and external cloud-controlling factors, in some degree, to escape the problems in tuning the cloud simulation (Klein et al., 2017; Mckim et al., 2024; Myers et al., 2021). The vertical gradient of the potential temperature ($d\theta/dz$), namely static stability, is the essential constraint of the environmental thermal stratification to the mass vertical transport. The changes in $d\theta/dz$ could modify the vertical distribution of the aerosols and water vapor and thus are the first-order cloud-controlling factor in all scales and play a critical role in the climate feedback processes (Jost and Robert, 2009; Li et al., 2014; Wilson Kemsley et al., 2024; Ceppi and Gregory, 2017; Bony et al., 2016; Koshiro et al., 2022; Bony et al., 2020).





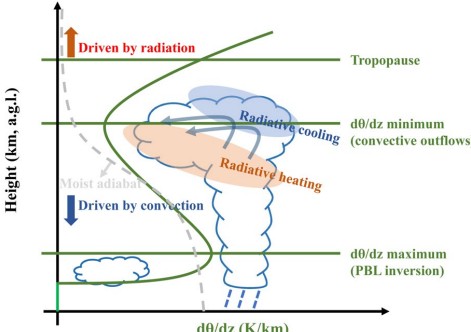

**Figure 1.** Illustration of the characteristics of the tropospheric stability profile and its relationship with clouds.

As illustrated in Fig. 1, it has been well observed and understood that the tropospheric $d\theta/dz$ profiles can be characterized by: (1) a maximum for the planetary boundary layer (PBL) capping inversion (Wood and Bretherton, 2006; Wang et al., 2023) and (2) a minimum atop the level of the convection outflow in respect of the tropical tropopause layer (TTL) convection-to-radiation transition (Birner et al., 2010; Gettelman and Forster, 2002). $d\theta/dz$ in the lower and middle troposphere is relatively well measured. At the PBL top, the inversion corresponds to the maximum of $d\theta/dz$ above the lifting condensation level (Wang et al., 2023). The inversion strength is the strongest low-cloud controlling factor by trapping the moisture below the PBL and inhibiting the upper-level dry cold air entering into the PBL (Bretherton and Wyant, 1997; Mohrmann et al., 2019; Klein and Hartmann, 1993; Wood, 2012). In the free troposphere, $d\theta/dz$ would be controlled by two critical processes: moist adiabat and baroclinic adjustment (Stone and Carlson, 1979). With the weak Coriolis force, tropical free-tropospheric $d\theta/dz$ is determined by the latent heat release (neutrally buoyant for the air-parcel ascending) and the observed values are normally within 20 % of the theoretical moist-adiabatic lapse rate (Held, 1982; Wood and Bretherton, 2006; Sobel et al., 2001; Stone and Carlson, 1979). Midlatitude free-tropospheric $d\theta/dz$ is more stable than that of the moist adiabat due to the meridional energy exchanges in the baroclinic waves and their values are proportional to the meridional $\theta$ gradients (Juckes, 2000; Frierson, 2008, 2006).

In the upper troposphere, the minimum of $d\theta/dz$ results from the TTL transition of two stability regimes, as illustrated in Fig. 1, from the convection-controlled troposphere ($d\theta/dz$ decreasing with the height) to the radiation-controlled stratosphere ($d\theta/dz$ increasing with the height) (Gettelman and Forster, 2002; Sunilkumar et al., 2017; Mehta et al., 2008). The $d\theta/dz$ values are determined by both the latent heat release and radiation and interactive with the anvil clouds. The top radiative cooling and the middle radiative and latent heating in the anvil would destabilize the cloud layer and this instability is the main cause of the spreading and thinning of anvil clouds (Gasparini et al., 2019; Lilly, 1988; Wall et al., 2020; Ruppert and Hohenegger, 2018; Hartmann et al., 2018; Ackerman et al., 1988).

In previous studies, the most widely-used metric for the upper-tropospheric stability (UTS) was defined as the $\theta$ difference of a 3-km deep layer below the lapse-rate tropopause (LRT) (Li et al., 2014; Wilson Kemsley et al., 2024). The LRT is defined by the World Meteorological Organization (WMO), i.e., the lowest level of the temperature lapse rate less than 2 K/km and the average lapse rate within 2 km above this level also less than 2 K/km. However, Tinney et al. (2022) argued that the WMO LRT definition fails to reliably identify the tropopause for the composition changes in the TTL but $d\theta/dz$ would be a superior metric to discriminate the TTL changes. Additionally, the WMO LRT height variation (normally around 16-17 km in tropics, Seidel et al. (2001) and Munchak and Pan (2014)) is more related to the ozone behavior but might be too high for anvil clouds (mostly distributed between 8-14 km, Yuan et al. (2011)). From tropics to the extratropics, the LRT transition is not continuous and the extratropics are normally characterized by multiple LRTs (Randel et al., 2007; Schmidt et al., 2006). It is not known whether in the extratropics the UTS under these complicated LRT conditions is still appropriate to represent the stability control on the anvil. If the level of the stability driving the anvil thinning and spreading could be captured from the $d\theta/dz$ profiles, a more skillful stability metric is possible. According to the fixed anvil-top temperature (FAT) hypothesis, anvil would be





produced at the altitude of the maximum divergence that is well consistent with the peak of the clear-sky radiative-driven convergence (Zelinka and Hartmann, 2010; Hartmann and Larson, 2002). The height of the dθ/dz minimum coincides with the anvil top of the maximum convective outflows (Gettelman and Forster, 2002; Mehta et al., 2008) and thereby could be more representative for deriving the stability control on the anvil than the LRT.

In this work, a novel stability metric in the upper troposphere is going to be proposed to better reflect the relationship between the stability and anvil clouds. This work aims to answer three questions:

(1) Based on the high-resolution radiosonde observations in the upper troposphere, what is the most appropriate stability metric of controlling the anvil clouds?

(2) To what extent can the stability metric be reproduced from the coarse-resolution reanalysis data?

(3) How well does the static stability control the anvil clouds?

This paper is structured as follows: Sect. 2 describes the data and processing methods; Sect. 3 introduces and evaluates the novel stability metric for controlling anvil clouds by ground-based atmospheric and cloud observations, to answer the first two questions; Sect. 4 presents the relationship between static stability and global anvil clouds by satellite observations, to answer the third question. Sect. 5 presents conclusions and answers to the three questions.

## 2. Data and methods

### 2.1 Radiosondes and cloud observations at the ground-based site

Ground-based observations of the 35-GHz millimeter cloud radar (MMCR) and balloon-borne radiosondes at Manus Island (147.4°E and 2.1°S, 2001-2011) are derived from the Atmospheric Radiation Measurement (ARM) Program. ARM was established by the US Department of Energy's Office of Biological and Environmental Research to provide observational basis to study the Earth climate. By the MMCR of the temporal and vertical sampling of 10 s and 45 m, cloud profiles over the site can be detected up to 20 km (S. Giangrande, 1999). The balloon-borne radiosondes are launched two times a day at around 11:30 and 23:30 UTC to provide atmospheric data with the temporal and vertical resolution of 2.5 s and 10 m (E. Keeler, 2001). The accuracy of the detected temperature and pressure is 0.2 K and 0.5 hPa, respectively. The θ and dθ/dz profiles can be easily computed according to the temperature and pressure profiles after the Gaussian 3-km-moving smoothing to avoid the noises. Based on the high vertical resolution of the radiosondes, the LRT and UTS can be computed strictly following their definitions (as presented in the introduction), respectively. The dθ/dz minimum height and values are derived from the dθ/dz profiles between 5-18 km.

The balloon usually takes hours to reach the upper troposphere. With the balloon drifting upward, the cloud detections by the MMCR within ± 5 minutes nearest to the balloon height are collected to compute the cloud fraction. The profiles of the ice cloud fraction are computed as the ratio of the number of ice cloud occurrence to the total number of observations at each height. Cloud layers are identified as the continuous levels of the cloud fraction greater than zeros with the distance less than 250 m. Clouds with the top temperature < -30 °C are identified as ice clouds. The cloud top height refers to the top height of the thickest ice cloud layer.

### 2.2 High cloud coverage (HCC) observations from radiometer imagers

Daily HCC between 60°S-60°N in 2007 passively observed by the radiometer imagers on board the geostationary satellites (GEOs) and the Moderate Resolution Imaging Spectroradiometer (MODIS) on board Aqua and Terra satellites is derived from the Synoptic 1° (SYN1deg) edition 4.1 product of the Clouds and the Earth's Radiant Energy System (CERES) Project (Doelling et al., 2013; Doelling et al., 2016). The HCC refers to the area fraction of the high clouds with the top pressure less than 300 hPa in 1-degree girds. The pixel-level high cloud identification of GEOs and MODIS is based on the similar CERES-MODIS cloud algorithm for uniform cloud observations (Trepte et al., 2019). The latest generation of GEO imagers with more additional channels is used to enhance the accuracy of cloud retrievals. In 2007, the second-generation GEOs of five channels (including 0.65, 3.9, 6.7, 11 and 12 μm) is used in the SYN1deg product to determine the cloud mask. In the CERES cloud algorithm, the cloud mask primarily depends on the cascading threshold tests (Minnis et al., 2008; Minnis et al.,



2011). The first check of the cloudy pixel is brightness temperature at 11 μm less than 260 K. For the thin cirrus that would not be detected by the first check, the radiance in terms of other channels is compared to the predicted clear-sky radiance to identify those thin cloud mask. Due to the limitation of the passive sensors, very thin cirrus of optical depths smaller than 0.3
would not be well detected by the CERES algorithm (Mace et al., 2005).

**2.3 Ice cloud vertical observations from the radar-lidar (DARDAR) measurements**

Instantaneous cloud profiles detected by both the Cloud Profiling Radar on board the Cloudsat satellite and the Cloud-Aerosol Lidar with Orthogonal Polarization instrument on board the Cloud-Aerosol Lidar and Infrared Pathfinder Satellite Observations (CALIPSO) are retrieved from the DARDAR project (Delanoë and Hogan, 2010, 2008). Radar can penetrate
thick clouds and lidar can detect very thin clouds. Their combination would provide more accurate full cloud vertical distributions. Additionally, radar and lidar is sensitive to large (ice) and small (liquid water) cloud particles, respectively, and thus their combination would benefit the discrimination of cloud phases. In the region of the temperature between -40-0 ℃, ice particles are discriminated from the supercooled droplets by the large radar reflectivity but small lidar backscatter signals (Delanoë and Hogan, 2010; Hogan et al., 2004).

**2.4 Atmospheric reanalysis**

The fifth-generation atmospheric reanalysis (ERA5) from the European Centre for Medium-Range Weather Forecasts (ECMWF) with the hourly and 0.5° resolution is used. The ERA5 profiles have 16 pressure levels between 50 and 500 hPa with the vertical resolution about 500 meters in the upper troposphere. Temperature, geopotential and divergence profiles are used in this work. The $dT/dz$ and $d\theta/dz$ profiles are computed from the ERA5 temperature and geopotential profiles at the half
levels. The method determining the WMO LRT is consistent with that proposed by Reichler et al. (2003). The half level of $-dT/dz$ less than 2K/km is first located between 5-18 km. The average 2-km-layer $-dT/dz$ above the half level is further computed. The lowest half level of $-dT/dz$ and the average 2-km-layer $-dT/dz$ both less than 2 K/km is found to linearly interpolate the exact position of the LRT. It has been verified that the root-mean-square errors of the reanalysis-based LRT is about 30-40 hPa in extratropics and 10-20 hPa in tropics in comparison to the radiosonde measures (Reichler et al., 2003; Meng et al., 2021).
The UTS is calculated according to the definition in Li et al. (2014), i.e., $d\theta/dz$ between the LRT and the level of 3 km below it by the linear interpolation from the ERA5 profiles. The ERA5 $d\theta/dz$ minimum height and values refers to that at the half levels between 5-18 km.

**2.5 Statistical methods**

The Pearson's correlation coefficient (R) and the slope of the least-squares linear fit are used. The existence of a
145 correlation is estimated based on the t-test. The number of independent samples is determined by dividing the total length of samples by the distance between independent samples (Bretherton et al., 1999). All correlations listed in this study are at the 95% significant level if without a mention of their significance. The confidence bound of R is computed based on the Fisher-Z Transformation.



### 3. Estimated anvil-top stability (EAS)

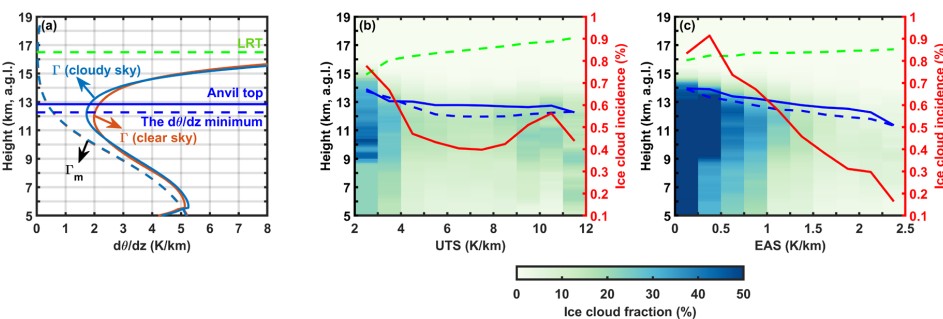

**Figure 2.** The thermal stratification in the upper troposphere and the controls of different stability metrics on the ice clouds at the ARM Manus site (147.4°E and 2.1°S) based on the radiosondes and radar observations from 2001 to 2011. (a) Composites of the environmental $d\theta/dz$ ($\Gamma$, solid blue and red lines) and the moist-adiabatic $d\theta/dz$ ($\Gamma_m$, dash blue and red lines) for cloudy and clear skies, respectively. The composites of the ice cloud fraction (cloud top temperature < -30℃) in each bin of the UTS (b) and EAS (c). The red line is the ice cloud incidence (the occurrence of the profiles with the ice cloud fraction greater than 0). The blue solid and dash lines indicate the mean anvil top and height of the $d\theta/dz$ minimum, respectively. The green dash line is the height of the LRT.

In Fig. 2a, with high-resolution radiosondes, the difference in the mean upper-tropospheric thermal stratification between cloudy and clear skies is examined at the ARM Tropical Western Pacific (TWP) Manus site. Here, the cloudy and clear sky refers to the existence of clouds above 5 km based on the ground-based MMCR. It is shown that from 6 to 11 km above the ground level (a.g.l.), observed environmental $d\theta/dz$ of either the cloudy or clear sky basically conforms to the moist adiabat, but is on average 0.46 K/km higher than the moist adiabatic $d\theta/dz$ possibly related to the absorption of water vapor to the longwave radiation. Similarly, Stone and Carlson (1979) also found that the mean lapse rate in the middle and lower troposphere is well close to the moist-adiabatic lapse rate with the difference no more than 0.4 K/km. The main difference in the mean $d\theta/dz$ profiles with the ice cloud existence above 5 km relative to that of the upper-tropospheric clear sky is the minimum of $d\theta/dz$ between 11-14 km. The mean height of the $d\theta/dz$ minimum (12.3 km) could correspond to the level of the maximum convective outflows (Mehta et al., 2008; Gettelman and Forster, 2002) and is also close to the anvil top (12.8 km) but a little lower as observed in Fig.1a. In contrast, the LRT is at 16.5 km and about 3.7 km higher than the anvil top on average. The UTS is computed from the 3-km-deep layer below the LRT but as shown in Fig. 1a from 14 km to the tropopause (16.5 km) the lapse rate is mainly contributed by the radiation processes due to the ozone without significant difference between the cloudy and clear sky. And thus, it is not expected that the variation in the UTS should be very closely related to the anvil. It seems that the minimum in $d\theta/dz$ could be used as the novel EAS index for two reasons: (1) the height of the minimum in $d\theta/dz$ is close to the anvil top (Fig. 1) of the maximum convective outflows (Mehta et al., 2008; Gettelman and Forster, 2002); (2) the variation of the $d\theta/dz$ minimum values is the most significant difference of the $d\theta/dz$ profiles between the cloudy and clear sky above 5 km.

The relationships of ice cloud incidence and fraction with the UTS based on the LRT and the novel EAS referring to the minimum stability in the upper troposphere are shown in Fig. 2b-c, respectively. When the UTS is small in Fig. 2b, the LRT (the green dash line) is relatively close to the anvil top (the blue solid line) and the small UTS also corresponds to the large ice cloud incidence and fraction. With the UTS larger, the LRT height increases but the anvil top decreases and their height difference increases from 1 km up to 5.2 km. And the ice cloud incidence decreases from 0.78 to 0.40 with significant fluctuations in the UTS bins of 7-12 K/km. It seems that in the bins of large UTS, the LRT is far away from the anvil and the UTS is not a strong controlling factor for anvil clouds. It is interesting to note that the height of the minimum in $d\theta/dz$ is always around the anvil top no matter what the UTS is in Fig. 2b, and thus the EAS could be a superior metric to control anvil clouds. It can be seen that the ice cloud incidence from 0.16 to 0.91 and its vertical fraction profiles are much better stratified by the EAS in Fig. 2c as compared to that stratified by the UTS in Fig. 2b. With the EAS increasing in Fig. 2c, the anvil top and the





height of the EAS are nearly coincident and both decrease from 14 to 11.3 km. It is physically reasonable that the convection
of lower EAS could overshoot to higher altitude and produce more anvil, while larger EAS would accelerate the anvil

dissipation.

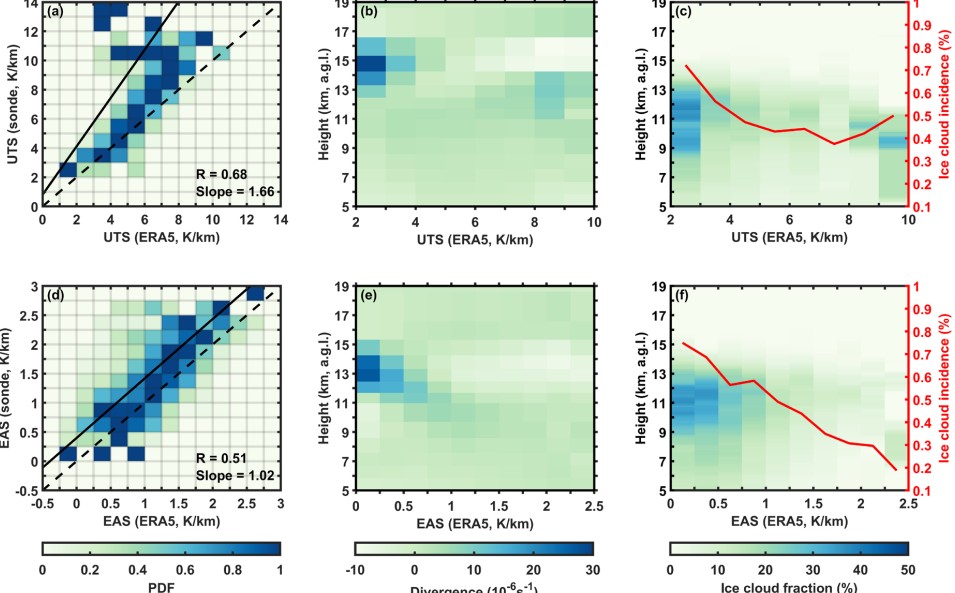

**Figure 3.** The accuracy of the ERA5-based stability metrics and their relationships with the divergence and ice clouds. (a) The
joint PDF between the radiosonde-measured and ERA5-based UTS (the PDF is normalized by dividing the maximum value).
(b-c) The relationship of the ERA5-based UTS with the D and ice cloud fraction, respectively. (d-f) Similar to (a-c) but for the

EAS. The red lines in (c) and (f) are the ice cloud incidence.

With the high-resolution radiosondes, the EAS is a more appropriate stability metric of controlling anvil clouds.
Nevertheless, the high-resolution radiosondes are limited at islands and coastal sites or during short-term field campaigns. To
pursue the general global and climate analyses, the EAS derived from the coarse-resolution atmospheric profiles of the

reanalysis is more applicable but needs further examinations about how well the EAS could be reproduced from the reanalysis.
In Fig. 3, the ERA5-based UTS and EAS is examined for their relationship with the radiosonde measures, divergence and ice
clouds, respectively. In Fig. 3a, the ERA5-based UTS is well correlated with the radiosonde-measured UTS with the correlation
of 0.68. The slope of the radiosonde-measure UTS to the ERA5-based UTS is 1.66 and thus the real UTS variations would be
underestimated by the coarse-resolution ERA5 data. Similar to Fig. 2b, the small and large UTS both could correspond to the

large divergence (Fig. 3b) and high incidence of ice clouds (Fig. 3c). This implies that the UTS would not well control the
convective divergence and anvil clouds. In Fig. 3d, the R between the ERA5-based and radiosonde-observed EAS is 0.51 and
their slope of the linear regression is 1, but the ERA5-based EAS is about 0.4 K/km lower than those radiosonde measures.
Fig. 3e shows that the ERA5-based EAS well sorts the variations of the strength and locations of the maximum divergence.
Less stable environment corresponds to the larger divergence at the higher altitude. In Fig. 3f, the ice cloud incidence decreases

from 0.75 to 0.19 with the ERA5-based EAS increasing from 0 to 2.5 K/km. In comparison to the Fig. 2c, the variation in the
ice cloud fraction profiles and incidence controlled by the ERA5-based EAS is relatively weaker than that by the radiosonde-
observed EAS. The ERA5-based EAS could not perfectly reproduce the effects of the real radiosonde-observed EAS on the
ice clouds due to the limitation of the coarse resolution. But it does better control the ice clouds than either the ERA5-bsaed
or the radiosonde-observed UTS.

**4.    On the relationship between the global EAS and anvil clouds**



The relationships of the ERA5-based stability metrics (i.e., the UTS and EAS) with global anvil clouds are further compared in this section. Here, the UTS and EAS are derived from the hourly ERA5 atmospheric profiles. It should be noted that the vertical location of the UTS and EAS is moving with time and thus they could not be captured from the long-term averaged profiles due to their variable locations in the profiles. Anvil clouds are detected in two ways by the passive GEO imagers and active DARDAR, respectively. The GEO imagers have the advantages of observing cloud spatial coverage and its continuous hourly variation. Active sensors of DARDAR can penetrate the cloud to provide the instantaneous cloud profiles but have sparse spatiotemporal sampling due to the sun-synchronous orbit. In this section, the controls of the ERA5-based stability metrics to the GEO-observed anvil spatial coverage and DARDAR-observed vertical profiles, respectively, are both examined.

The primary difference between the UTS and EAS is the height of calculating the static stability. The UTS is based on the WMO-defined LRT, while the EAS is based on the height of the minimum $d\theta/dz$. In Fig. 4a, the ERA5-based LRT is uniform in tropics within 16-17 km and decreases rapidly to about 8 km in the midlatitude, which is well consistent with the LRT observed by the radiosondes in Seidel et al. (2001) and the global positioning system (GPS) radio occultation (RO) in Rieckh et al. (2014). In Fig. 4b, the height of the minimum $d\theta/dz$ from the ERA5 profiles in the deep tropics is about 10-13 km, and is consistent with the radiosonde and GPS RO observations in Gettelman and Forster (2002) and Sunilkumar et al. (2017). The height of the minimum $d\theta/dz$ well corresponds to the height of the divergence prevailing above 10 km (Mapes and Houze, 1995) and the anvil dominating about 8-14 km (Yuan et al., 2011; Yuan and Houze, 2010). Gettelman and Forster (2002) also indicated that the height of the minimum $d\theta/dz$ is closely associated with the cloud height and divergence of the wind field. This approves that the height of the minimum $d\theta/dz$ would be more reasonable than the LRT for calculating the stability of controlling the anvil clouds.

In Fig. 4c-e, the annual mean geographic distributions of the ERA5-based UTS, EAS and GEO-observed HCC are shown, respectively. The UTS is low in both the deep tropics and midlatitude but has a strong high-stability belt in the subtropics, which is consistent with that in Li et al. (2014). The EAS is less stable in tropics and the most unstable in the Tropical West Pacific and Tibet Plateau (Fig. 4d) corresponding to the largest HCC (Fig. 4e). Toward the high latitude, the EAS (HCC) gradually increases (decreases). In terms of their spatial distributions, the EAS is well correlated with the HCC with R of -0.60 at 95 % significant level, while the UTS is barely correlated with the HCC distribution of R only -0.01.




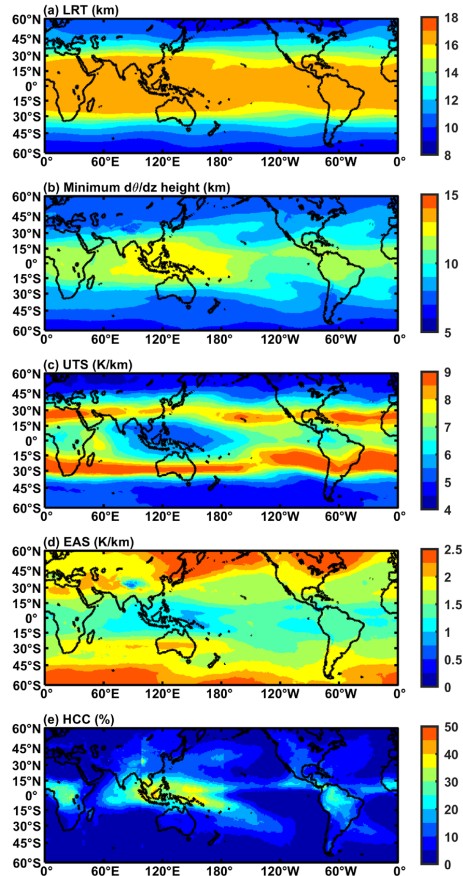

**Figure 4**. Geographic distribution of the ERA5-based (a) LRT, (b) minimum dθ/dz height, (c) UTS, (d) EAS and (e) the GEO-osberved HCC.

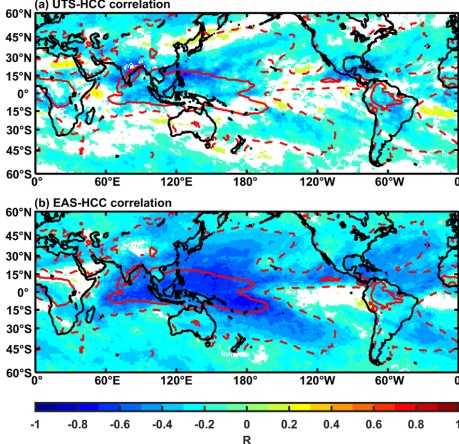

**Figure 5.** Relationship of daily GEO-observed HCC with the ERA5-based UTS and EAS. (a) Correlation between the daily-averaged HCC and UTS. (b) Correlation between the daily-averaged HCC and EAS. Only correlation at 95 % significance level is shown. The red solid and dash lines indicate the contours of HCC 20 % and 5 %, respectively.



In Fig. 5, the correlation of the daily-averaged UTS and EAS with the HCC is shown. The variations of the HCC are more correlated with the EAS as compared with the HCC-UTS correlation. The HCC could not be significantly correlated with the UTS even in the high cloud domains and the HCC correlation with the UTS is much smaller than that with the EAS. On average, the UTS-HCC correlation over the domains of the HCC larger than 5 % and 20 % is -0.26 and -0.33, in which only the correlation at the 95 % significant level is counted. The averaged EAS-HCC correlation over the 5 % and 20 % HCC domains is -0.40 and -0.52, respectively. Especially over the tropical western Pacific, most of the EAS-HCC correlation is within -0.6 to -0.7. It is interesting to note that the stability is a very strong anvil-controlling factor but this has not been noticed before since the UTS underestimates the stability control on the anvil clouds. In fact, the EAS-HCC correlation is comparable to the correlation between the inversion (the strongest low cloud controlling factor) and low clouds (about 0.6 in the low cloud domains in Wang et al. (2023)). By the observational constraints of low cloud controlling factors, Myers et al. (2021) have successfully reduced the uncertainty of the low cloud climate feedbacks in the climate models. Given their successes, as a good predictor of the anvil, the EAS could be helpful to understand and predict the anvil climate feedbacks.

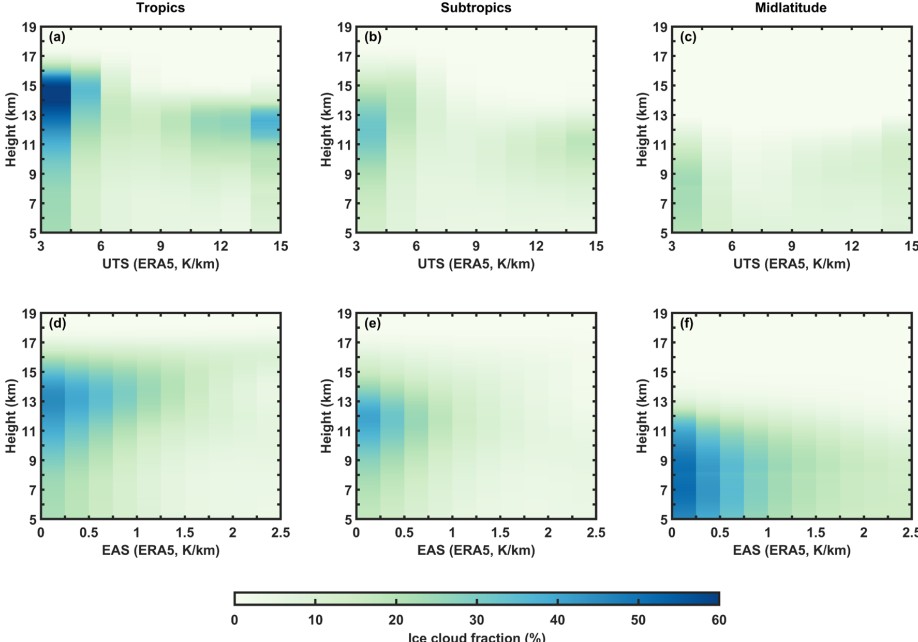

**Figure 6.** Relationship between the DARDAR-detected ice cloud fraction profiles and the ERA5-based UTS and EAS over oceans. (a-c) The composited ice cloud fraction in each bin of the UTS over tropics, subtropics and midlatitude, respectively. (d-f) The composited ice cloud fraction in each bin of the EAS over tropics, subtropics and midlatitude, respectively.




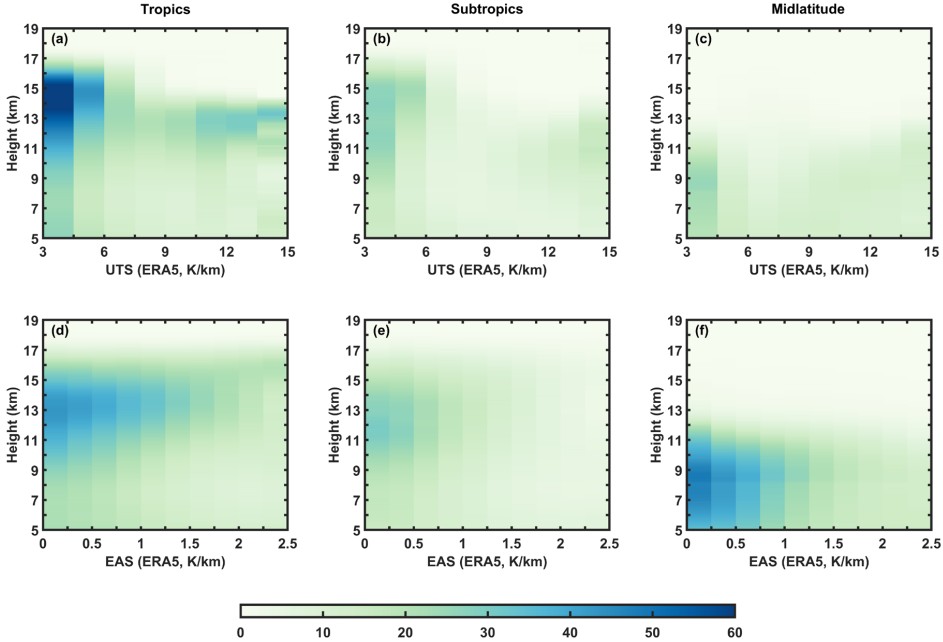

**Figure 7.** Similar to Fig. 6 but over the land.

The relationship of the instantaneous ice cloud profiles detected by the DARDAR with the ERA5-based stability metrics is shown over oceans (Fig. 6) and land (Fig. 7), respectively. Over oceans, two modes are found for the relationship between the ice cloud fraction and UTS in either tropics (Fig. 6a), subtropics (Fig. 6b) or midlatitude (Fig. 6c). For the UTS within 3-9 K/km, the UTS is negatively correlated with the ice cloud fraction, while for the UTS within 9-15 K/km the UTS has a positive correlation with the ice cloud fraction. This is similar to the result in Fig. 2b for the Manus site. The negative correlation
between the UTS and ice cloud fraction is understandable since the stability is inversely proportional to the convective outflows, but why the positive correlation for the large UTS happens is not known. Possibly, larger UTS usually corresponds to higher LRT but farther away from the anvil and thus gradually loses the control to the anvil. In contrast, the ice cloud fraction is well linearly sorted by the EAS in either the tropics (Fig. 6d), subtropics (Fig. 6e) or midlatitude (Fig. 6f). The relationship between the ice cloud fraction and EAS is more physically reasonable that more unstable upper-tropospheric environment would favor
stronger convective outflows and more sustain the anvil formation. Similarly, over the land, two modes of the relationship of the ice cloud fraction with the UTS are found in Fig. 7a-c, while the linear relationship of the ice clouds with the EAS is shown in Fig. 7d-f. Fig. 6 and Fig. 7 show that the controls of the stability on the ice cloud fraction over oceans and land are nearly the same.

        Overall, the ERA5-based EAS is better correlated with the anvil clouds than the UTS. As observed by the GEO (Fig. 4-
5), the EAS better explain the HCC spatial distribution and its temporal variation than the UTS. As observed by the DARDAR (Fig. 6-7), the EAS more linearly constrains the ice cloud fraction than the UTS.

**5. Conclusions**

        As a novel stability metric, the EAS is proposed for representing the stability control on the anvil clouds. In comparison with the UTS based on the LRT, the EAS refers to the minimum static stability in the upper troposphere. Grounded on the
observations of high-resolution radiosondes and MMCR at the ARM Manus site, the height of the dθ/dz minimum (EAS) is close to the anvil top of 0.5-km difference, while for the UTS the LRT is normally 3.7 km above the anvil top. The closer distance of the EAS with the anvil would enable the EAS to better represent the environmental stability of controlling the anvil.



Similarly, Gettelman and Forster (2002) and Mehta et al. (2008) also approved that the height of the dθ/dz minimum is closely associated the cloud height and corresponds to the height of the maximum convection outflows. In addition, the results at the Manus site show that the amplitude of the dθ/dz minimum (EAS) is the most distinguishable feature of the observed stability profiles between the ice-cloudy and clear sky. With the cloud observations by the Manus MMCR, the EAS has a strong negative linear correlation with the ice cloud occurrence (from 0.16-0.91) and its vertical fraction profiles, while small (large) UTS is negatively (positively) correlated with the ice cloud occurrence of a narrow range within 0.4-0.78. This approves that the EAS can better represent the stability controls on the anvil than the UTS.

The ERA5 pressure-level atmospheric profiles are more appliable than the radiosondes at the global and climate scales, while the ERA5 profiles have relatively coarse vertical resolution about 500 m in the upper troposphere. The results show that the ERA5-based EAS would be 0.4 K/km smaller than the radiosonde-measured EAS with the linear correlation of 0.51 and the slope of nearly 1. For the controlling effects on the anvil, the ERA5-based EAS is still negatively correlated with the ice cloud occurrence but its range resolved by the ERA5-based EAS (0.19-0.75) is smaller than that by the radiosonde-measured EAS (0.16-0.91). Nevertheless, the ERA5-based EAS still servers as a stronger anvil-controlling factor than either the radiosonde-measured or ERA5-based UTS.

At the global scales, the stability controls on the anvil revealed by the ERA5-based EAS are much stronger than that by the UTS. As observed by the GEO, the HCC-EAS spatial correlation is -0.6, while the HCC spatial distribution is barely correlated with the UTS with the correlation of -0.01. For the temporal variations, the daily HCC is more correlated with the EAS than the UTS, with the correlation of -0.52 and -0.33 over the HCC domains, respectively. Especially over the tropical western Pacific, most of the daily HCC-EAS correlation is as strong as -0.6. Additionally, as observed by the active sensors of the DARDAR on board the A-Train satellites, the ice cloud fraction profiles are all more linearly stratified by the EAS as compared with that by the UTS. The results suggest that the EAS is a strong anvil-controlling factor in all the tropics, subtropics and midlatitude and the controlling effects over oceans and land have no significant difference. It is interesting to note that the observed linear relationship between the ERA5-based EAS and the anvil clouds is not worse than the relationship between the inversion (the strongest low-cloud controlling factor) and low clouds. This implies that the EAS would be helpful to understand and predict the anvil climate feedbacks and could be used as the observational constraint to reduce the model uncertainty.

**Acknowledgment**

This work was supported by the NSFC-41875004 and the National Key R&D Program of China (2016YFC0202000).

**Author contribution**

ZW designed and completed this work.

**Data and code availability**

All data used in this study are available online. The ground-based cloud (S. Giangrande, 1999) and radiosonde (E. Keeler, 2001) observations at the ground-based sites are obtained from the Atmospheric Radiation Measurement user facility, a U.S. Department Of Energy (DOE) office of science user facility managed by the biological and environmental research program (https://www.arm.gov). The CERES SYN1deg product (Nasa/Larc/Sd/Asdc, 2017) are obtained from the National Aeronautics and Space Administration (NASA) Langley Research Center Atmospheric Science Data Center (https://search.earthdata.nasa.gov/). The DARDAR product (Delanoë and Hogan, 2008) is available at https://www.icare.univ-lille.fr/dardar/. The ERA5 (Hersbach, 2023) is obtained from the Climate Data Store (https://climate.copernicus.eu). The code of calculating the reanalysis-based EAS is available upon request.

**Competing interests**

The author declares that he has no conflict of interest.

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
