# Peer review of "On the relationship between static stability and anvil clouds"

_EGUsphere, 2024_

## Referee Comment (RC1)

**Comments on "On the relationship between static stability and anvil clouds" by Zhenquan Wang**

This paper presents a statistical analysis of the relationship between high cloud fraction and upper-level static stability, as measured by two different metrics. The main conclusion is that the newly proposed stability metric (EAS) has a stronger relationship to ice cloud fraction than a previously used metric (UTS). This result is supported by the analysis, and the author suggests it is useful for studies that use "cloud-controlling factor" analysis. The analysis itself seems sound, although there are some aspects that can be clarified.

I find the motivation of this study and the usefulness of its main result to be unclear. There are also some significant flaws in the presentation of previous work in this area, and as a result the physical interpretations are not always sound. At the end of the day, this paper provides an incremental improvement to how one specific cloud-controlling factor might be calculated.

**Paper Organization and Writing**

- In my opinion, the use of the term "anvil clouds" is misleading. The analysis includes all high ice clouds. In the tropics, many of these may be convectively generated anvils, but a large portion are thin cirrus clouds formed in situ. In the midlatitudes and subtropics, depending on the time of year, the upper-level ice clouds are dominated by midlatitude weather systems. From what I can tell, there is no attempt in the paper to distinguish anvil clouds from other ice clouds. "High clouds" or "High ice clouds" may be more appropriate.

- I found the Introduction to be a bit off-topic at times. There is ample discussion of low clouds, which are not the subject of this paper, yet little discussion of *why* there is a relationship between stability and high clouds in the first place. In addition, some previous work on anvil clouds is misrepresented (details below). I suggest the author refocus the Introduction on the use of stability as a high-cloud controlling factor. The existing discussion of this topic, and the discussion of LRT, are most interesting and useful.

- There are many spelling, syntax, and vocabulary errors in this paper. I have listed a few of them in the line comments. I recommend that the author pursue professional editing help if such services are available at the author's institution.

**General comments on the study**

- **Conflation of various relationships & hypotheses regarding anvil clouds and static stability.** The author touches on a few different ideas involving anvil clouds and stability. As I see it, these can be separated into
    - **(1)** The use of an upper-level stability metric in cloud-controlling factor analysis of high cloud amount, such as in Li et al (2014).

**(2)** The relationship between stability, clear-sky convergence, and anvil cloud fraction as laid out in Zelinka & Hartmann (2010, doi:10.1029/2010JD013817) and reformulated into the Stability Iris hypothesis by Bony et al (2016).

**(3)** A process-level relationship between environmental stability and the evolution (i.e., lifetime) of detrained anvil clouds.

Of course, there is overlap between these ideas. But this study seems most relevant to (1), as it presents a new metric of upper-level stability that it claims to be more tightly linked to high cloud area. Ideas (2) and (3) are primarily used as motivation or to briefly speculate some reasons for the study results, but they are not always invoked correctly.

With regard to (2): this relationship between stability, clear-sky convergence (i.e. cloudy-sky divergence), and anvil cloud fraction is an equilibrium argument. The Stability Iris idea is about differences in stability between different climate states, not about differences in stability between cloudy and clear regions. In fact, the hypothesis assumes weak temperature gradients (i.e., minimal spatial variation in stability). Fig 3 is used to argue that low stability ->  divergence -> anvil clouds, while high stability -> convergence -> no clouds. This may be true, but it does not seem like a new finding. Deep convection will not penetrate strongly stratified areas of the Tropics, so it is preferentially occurs where there is lower stability…the upper-level divergence at low stability is required by mass continuity…and the clouds will of course be found where convection is. These spatial contrasts between cloudy and clear-sky regions are *not* the stability changes addressed by the Stability Iris ideas, and I don't see a clear connection to climate feedbacks (the connection is not necessary, but the authors use it as a motivation)

With regard to (3), there seems to be a misunderstanding of some previous research about anvil cloud evolution. The author suggests throughout the paper that greater ambient stability is associated with shorter anvil cloud lifetime, but this is not what the cited papers show. The most appropriate citation by the author may be Lilly (1988) (line 60). In Lilly's model, the stratification of the environment acts to flatten and spread the neutrally buoyant anvil—but this is a simple theoretical model that, to my knowledge, has never been tested. The most problematic misunderstanding is on lines 58-61, which cites some previous work about the vertical gradients of diabatic *heating* within anvils. But these hypotheses also assume weak temperature gradients, so the heating is balanced by vertical motion. This may lead to in-cloud mixing but does not actually affect the vertical temperature gradient. Moreover, the vertical gradient in heating has not been shown to be the main cause of anvil spreading and thinning. Wall et al (2020) and Gasparini et al (2022) suggested that diabatic heating and lofting of the entire layer, as opposed to the vertical gradient, is very important to the anvil life cycle. While this question is still unsettled, the in-cloud mixing played a lesser role in these studies.

- **Use of ground-based radar.** The MMCR is not the ideal choice of instrument for detecting cloud top height (CTH). The radar is not very sensitive to small ice crystals, which tend to dominate the upper parts of anvil cirrus. This can make a big difference in retrieved CTH and cloud fraction statistics; see the Key Figures on this

page: https://climatedataguide.ucar.edu/climate-data/combined-cloudsat-spaceborne-radar-and-calipso-spaceborne-lidar-cloud-fraction-dataset. As a result, the typical CTHs shown in Fig 2 are 1-2 km lower than previous studies using spaceborne lidar, e.g., Berry & Mace (2014; doi: 10.1002/2014JD021458), Hartmann & Berry (2017; doi:10.1002/2017JD026460), Dessler et al (2006; doi:10.1029/2005JD006705), among many others. The same bias can be seen when compared to the DARDAR cloud fraction in Fig 6, where CTHs look close to 15 km.

In addition, the MMCR can become attenuated by precipitation and optically thick clouds (Hollars et al 2004, doi:10.1016/j.atmosres.2004.03.015), which would also bias CTH low in the case of precipitating convection.

This issue does not necessarily invalidate the rest of the analysis, since this is not a study of CTH itself. But it means that UTS, which measures stability between ~13.5-16.5 km, does coincide rather closely with true CTH. This weakens the author's argument that EAS is favorable over UTS because of its closer proximity to the anvil. Nevertheless, the EAS-HCC relationship still seems to be better than UTS-HCC, so the main conclusion is not affected.

- Please specify how the moist adiabatic $d\theta/dz$ is calculated. Does the author take the observed pressure and temperature at some vertical level and use it calculate the moist adiabatic lapse rate? Or is it found by launching a moist adiabatic parcel profile from some assumed surface conditions? These two methods would give different results. Also, does the calculation use the saturation vapor pressure over liquid throughout the entire troposphere, or is there a transition to ice saturation at cold temperatures?

**Line Comments**

- Line 28: I found the wording "Cloud responses to the environmental changes have not been correctly simulated in models" to be a bit odd. We do not know if cloud responses have been correctly simulated, since we do not know the ground truth in future climate scenarios. I'd suggest more precise language for the first sentence of the paper, i.e., something about uncertainty.

- Line 35 "to the mass" -> "on the mass"

- Line 36-37: "are the first-order cloud-controlling factor in all scales"….this strikes me as an enormous claim that I have never heard before. *All* scales, even microphysical? Again, more precise language is needed here.

- Line 44-45 "convection-to-radiation transition" – again, not very precise language. I know what the authors mean here, but many readers may not.

- Line 49: "moist adiabat" -> "moist adiabatic ascent"

- Line 51: "(neutrally buoyant for the air-parcel ascending)" what do the authors mean by this? Is the author trying to say that ascending parcels are neutrally buoyancy? If so, that is not truly the case, just an approximation used in theoretical studies of the lapse rate, e.g. Singh & O'Gorman 2013.

- Line 59: "latent heat release and radiation and interactive with the anvil clouds" -> fix grammar

- Line 104: "with the distance less than 250" – specify that this is the clear-sky distance.

- Line 126: Lidar is sensitive to small ice particles in addition to liquid. As written, it sounds like radar detects only ice and lidar detects only liquid. DARDAR is an ice-only product that still relies heavily on the lidar.

- Line 133: shouldn't the vertical resolution be in hPa, not meters?

- Line 137: Sentence starting with "The lowest half level…" is unclear. In addition, I think the same symbol is being used to indicate a hyphen as well as a negative sign, which is confusing in this case

- Line 144: "existence" -> "significance"

- Line 146: "The number of independent samples is determined…by the distance between independent samples." Something is off here—you can't use the number of independent samples to determine the number of independent samples. And is this distance referring to spatial or temporal distance? If it is temporal distance, shouldn't the autocorrelation be used to determine the number of independent samples?

- Line 154: caption references a dashed red line but there is no dashed red line

- Line 164: absorption *of* longwave radiation *by* water vapor.

- Line 175: "is close to the anvil top of the maximum convective outflows"…is the author referencing the anvil top height or the height of max outflow? As they noted earlier, these heights are different.

- Fig 3: where are the results of the statistical significance tests for the correlations?

- Fig 3: Do pabels b, c, e, f show results just for the Manus location, or for the whole 60S-60N study region? It is sometimes hard to follow which instruments are being used in each of the figures (e.g., is ice cloud fraction still from MMCR or now from MODIS). It would be helpful to specify in the caption.

- Line 245: "approves" is not the right word here. Perhaps something like "supports the idea that"

- Line 254: what does the author mean by "in which only the correlation at the 95% significance level is counted"? Is the daily mean UTS and HCC being computed for the entire HCC regions, and then the correlations are computed using single values for the entire region? Or, are the daily mean correlations being computed for each grid point, and then the R values averaged across all grid points in the region? If it is the latter, I think all R values should be included in the averaging, not just those that meet the significance test.

- Line 276: could it be that large UTS forces convection to detrain at lower altitudes, producing more clouds in the 11-13 km range?

- Line 280: why would a more unstable environment sustain anvil clouds over time?
- Line 286: is this relationship indeed linear? This is hard to tell from the color scale used in Figs 6-7.
- Line 293: "approved" -> showed
- Line 298: "approves" -> shows
- Line 306: servers -> serves

---

## Referee Comment (RC2)

Review report on "On the relationship between static stability and anvil clouds" by Zhenquan Wang.

In this manuscript, the author tried to explore the relationship between high-level anvil clouds and static stability through a novel approach known as estimated anvil-top stability (EAS), which is based on the minimum value of $d\theta/dz$. The author claims that the upper-tropospheric stability (UTS) method, which relies on the lapse rate tropopause, underestimates the effect of stability on the anvil, while EAS provides a more accurate relationship. The findings further indicate that EAS has a stronger correlation with anvil clouds than UTS. This proposed method may be useful in understanding the factors that control cloud structure and composition. The topic of the research is interesting and the analysis results would be worth a concise publication. Though the manuscript is scientifically sound enough, the presentation style needs to be improved. Overall, the manuscript requires major revisions. I had the chance to read the comments of Anonymous Reviewer #2 and I do share all his/her general comments.

**Comments:**

1. What I can see as one of the major problems of the manuscript is that it lacks clarity in many places in its current form. Several sentences are not clear, please revise.
2. Is it 3 km moving smoothing? What is the basis for selecting 3-km smoothing? What is the final vertical resolution of radiosonde data?
3. Is the LRT derived from this 3 km smoothed temperature profile as well? What is the reliability of the relationship established by the results obtained?
4. The author may clearly describe how the Anvil cloud is categorized in this manuscript. What are the limitations of the MMCR for detecting the anvil clouds?
5. What is the time frame for the ERA-5 data and other satellite measurements? Does it align with the radiosonde data from 2001 to 2011? The author should provide clarification on this matter.
6. Similarly, the author needs to provide the spatial gridding of each data set in the study. What is the spatial resolution for ERA-5, CERES, and DARDAR? While using multiple data sets of observation, reanalysis, and satellite data, it is suggested that it should be gridded to a uniform resolution for better comparison.
7. What was the horizontal drift of the balloon while comparing the cloud fraction of MMCR?
8. How are the lower-level thin cirrus ice crystals accounted for if the ice clouds are identified based on cloud top temperature?
9. The methodology to estimate the moist adiabatic from observation and model datasets used in this study may be explained.
10. The height of the minimum potential temperature gradient (**Fig. 4b**), commonly known as the convective outflow level or convective tropopause, has been extensively studied and documented by numerous researchers and needs to be included and discussed in the present study (see the reference).

**Technical corrections:**

The English language used in the manuscript needs to be checked by professionals who are native English speakers.

Line 28: "Cloud responses to the environmental changes have not been correctly simulated in models" may be rewritten avoiding concluding statements.

Line 197: 'Nevertheless, the high-resolution radiosondes are limited at islands and coastal sites or during short-term field campaigns.' What about using the available high-resolution GNSS RO data? see the attached references.

Line 225: How is the ERA-5 data on pressure levels used to identify the height of LRT?

Figure 1 The tick labels are missing on the y-axis.

**References:**

Biondi, R., W. J. Randel, S.-P. Ho, T. Neubert, and S. Syndergaard, 2012: Thermal structure of intense convective clouds derived from GPS radio occultations. Atmos. Chem. Phys., 12, 5309–5318, https://doi.org/10.5194/acp-12-5309-2012.

Sunilkumar, S.V., Babu, A., Parameswaran, K., 2013. Mean structure of the tropical tropopause and its variability over the Indian longitude sector. Clim. Dyn. 40, 1125–1140. https://doi.org/10.1007/s00382-012-1496-8.

Ravindra Babu, S. "Convective Tropopause Over the Tropics: Climatology, Seasonality, and Inter-Annual Variability Inferred from Long-Term FORMOSAT-3/COSMIC-1 RO Data," Atmospheric Research, vol. 298, https://doi.org/10.1016/j.atmosres.2023.107159, 2024.

Ho, S.-P., and Coauthors, 2020: The COSMIC/FORMOSAT-3 radio occultation mission after 12 years: Accomplishments, remaining challenges, and potential impacts of COSMIC-2. Bull. Amer. Meteor. Soc., 101, E1107–E1136, https://doi.org/10.1175/BAMS-D-18-0290.1.

Xian, T., and Y. Fu, 2015: Characteristics of tropopause-penetrating convection determined by TRMM and COSMIC GPS radio occultation measurements. J. Geophys. Res. Atmos., 120, 7006–7024, https://doi.org/10.1002/2014JD022633.

---

## Author Comment (AC1)

**Response to Anonymous Referee #1**

*Referee #1: This paper presents a statistical analysis of the relationship between high cloud fraction and upper-level static stability, as measured by two different metrics. The main conclusion is that the newly proposed stability metric (EAS) has a stronger relationship to ice cloud fraction than a previously used metric (UTS). This result is supported by the analysis, and the author suggests it is useful for studies that use "cloud-controlling factor" analysis. The analysis itself seems sound, although there are some aspects that can be clarified.*

*I find the motivation of this study and the usefulness of its main result to be unclear. There are also some significant flaws in the presentation of previous work in this area, and as a result the physical interpretations are not always sound. At the end of the day, this paper provides an incremental improvement to how one specific cloud-controlling factor might be calculated.*

==Response:== We thank the anonymous reviewer for his/her efforts of reviewing our manuscript. We are very grateful for his/her valuable and insightful comments to help us to improve the interpretations of the results. We have carefully taken these comments into account, and accordingly revised the motivation and analyses.

**Paper Organization and Writing**

- *In my opinion, the use of the term "anvil clouds" is misleading. The analysis includes all high ice clouds. In the tropics, many of these may be convectively generated anvils, but a large portion are thin cirrus clouds formed in situ. In the midlatitudes and subtropics, depending on the time of year, the upper-level ice clouds are dominated by midlatitude weather systems. From what I can tell, there is no attempt in the paper to distinguish anvil clouds from other ice clouds. "High clouds" or "High ice clouds" may be more appropriate.*

  ==Response:== As suggested, the term "anvil clouds" has been replaced with "high ice clouds" in the revised manuscript.

- *I found the Introduction to be a bit off-topic at times. There is ample discussion of low clouds, which are not the subject of this paper, yet little discussion of why there is a relationship between stability and high clouds in the first place. In addition, some previous work on anvil clouds is misrepresented (details below). I suggest the author refocus the Introduction on the use of stability as a high-cloud controlling factor. The existing discussion of this topic, and the discussion of LRT, are most interesting and useful.*

  ==Response:== The discussions about the low-level stability have been removed. More discussions of why there is a relationship between stability and high clouds have been added on the basis of the following comments. Those misrepresentations of previous works have been corrected.

- *There are many spelling, syntax, and vocabulary errors in this paper. I have listed a few of them in*

*the line comments. I recommend that the author pursue professional editing help if such services are available at the author's institution.*

==Response:== Thanks a lot. We have carefully corrected grammar errors and pursued the professional editing help.

*General comments on the study*

- **Conflation of various relationships & hypotheses regarding anvil clouds and static stability.** *The author touches on a few different ideas involving anvil clouds and stability. As I see it, these can be separated into*
    - **(1)** *The use of an upper-level stability metric in cloud-controlling factor analysis of high cloud amount, such as in Li et al (2014).*
    - **(2)** *The relationship between stability, clear-sky convergence, and anvil cloud fraction as laid out in Zelinka & Hartmann (2010, doi:10.1029/2010JD013817) and reformulated into the Stability Iris hypothesis by Bony et al (2016).*
    - **(3)** *A process-level relationship between environmental stability and the evolution (i.e., lifetime) of detrained anvil clouds.*

*Of course, there is overlap between these ideas. But this study seems most relevant to (1), as it presents a new metric of upper-level stability that it claims to be more tightly linked to high cloud area. Ideas (2) and (3) are primarily used as motivation or to briefly speculate some reasons for the study results, but they are not always invoked correctly.*

*With regard to (2): this relationship between stability, clear-sky convergence (i.e. cloudy-sky divergence), and anvil cloud fraction is an equilibrium argument. The Stability Iris idea is about differences in stability between different climate states, not about differences in stability between cloudy and clear regions. In fact, the hypothesis assumes weak temperature gradients (i.e., minimal spatial variation in stability). Fig 3 is used to argue that low stability -> divergence -> anvil clouds, while high stability -> convergence -> no clouds. This may be true, but it does not seem like a new finding. Deep convection will not penetrate strongly stratified areas of the Tropics, so it is preferentially occurs where there is lower stability...the upper- level divergence at low stability is required by mass continuity...and the clouds will of course be found where convection is. These spatial contrasts between cloudy and clear-sky regions are not the stability changes addressed by the Stability Iris ideas, and I don't see a clear connection to climate feedbacks (the connection is not necessary, but the authors use it as a motivation)*

*With regard to (3), there seems to be a misunderstanding of some previous research about anvil cloud evolution. The author suggests throughout the paper that greater ambient stability is associated with shorter anvil cloud lifetime, but this is not what the cited papers show. The most appropriate citation by the author may be Lilly (1988) (line 60). In Lilly's model, the stratification of the environment acts to flatten and spread the neutrally buoyant anvil—but this is a simple theoretical model that, to my knowledge, has never been tested. The most problematic*

*misunderstanding is on lines 58-61, which cites some previous work about the vertical gradients of diabatic heating within anvils. But these hypotheses also assume weak temperature gradients, so the heating is balanced by vertical motion. This may lead to in-cloud mixing but does not actually affect the vertical temperature gradient. Moreover, the vertical gradient in heating has not been shown to be the main cause of anvil spreading and thinning. Wall et al (2020) and Gasparini et al (2022) suggested that diabatic heating and lofting of the entire layer, as opposed to the vertical gradient, is very important to the anvil life cycle. While this question is still unsettled, the in-cloud mixing played a lesser role in these studies.*

Response: Thank you for providing this framework. It is very clear and helps a lot. The ideas (2) and (3) are used as motivations to discuss how the stability influences anvil clouds, as a part of the introduction in the revised manuscript. The idea (1) is the main fucus of this work and is discussed for why and how a better stability metric of controlling anvil clouds is possible, in the introduction and a newly added section 3 in the revised manuscript. Misunderstandings and misrepresentations of previous works have been corrected.

The ideas (2) and (3) have been reorganized in the introduction part as: "The $d\theta/dz$ values are determined by both the latent heat release and the radiation, and the stability interacts with high ice clouds. On the process level, Lilly (1988) proposed a model to explain cirrus outflow dynamics, in which the initial outflowing cirrus clouds from convection are flattened and spread by environmental stability and are further maintained by the destabilization of radiative heating in the vertical direction. However, validating this hypothesized process has remained a challenge until now. Wall et al. (2020) and Gasparini et al. (2022) suggested that the horizontal gradients of radiative heating between anvil clouds and its adjacent environment, as opposed to the vertical in-cloud radiative destabilization, are very important for anvil spreading and thinning. Additionally, in climate studies, there is a physical link among static stability, clear-sky convergence and anvil cloud fractions according to radiative-convective equilibrium. As the climate warms, the mean state of the tropical atmosphere becomes more stable (Bony et al., 2016). This increase in stability requires weaker clear-sky diabatic subsidence and less convergence to maintain the energy balance, and ultimately leads to a shrink of convective regions and a reduction in anvil coverage, namely the Stability Iris (Zelinka and Hartmann, 2010; Bony et al., 2016). Overall, stability can be a strong controlling factor for high ice clouds. However, a representative metric for the static stability of controlling high ice clouds is lacking.

In previous studies, the upper-tropospheric stability (UTS) was defined as the $\theta$ difference in a 3-km-deep layer below the lapse-rate tropopause (LRT) (Li et al., 2014; Hong and Di Girolamo, 2020; Maleska et al., 2020; Wilson Kemsley et al., 2024). The LRT is defined by the World Meteorological Organization (WMO), which requires (1) the lowest level of the temperature lapse rate is less than 2 K/km, and (2) the average lapse rate within 2 km above this level is also less than 2 K/km. However, Tinney et al. (2022) argued that the WMO LRT definition fails to reliably identify the tropopause for the composition changes in the TTL, but $d\theta/dz$ would be a superior metric to discriminate the TTL changes. Additionally, the LRT height variation (normally 16-17 km in the tropics, Seidel et al. (2001) and Munchak and Pan (2014)) is more related to ozone behaviors and radiative processes, but might be too

high for dynamical processes, such as anvil clouds (mostly distributed between 8-14 km, Yuan et al. (2011)) and convective outflows (~10-12 km, Mapes and Houze (1995) and Folkins (2002)). Similarly, Highwood and Hoskins (1998) also argued that the LRT is rather arbitrarily defined but has limited physical relevance, and there seems to be little connection between any convective processes and the lapse-rate definition. From the tropics to the extratropics, the LRT transition is not continuous, and the extratropics are normally characterized by multiple LRTs (Schmidt et al., 2006; Randel et al., 2007). In the extratropics under these complicated LRT conditions, it is not known whether UTS represents the stability control on high ice clouds appropriately.".

The idea (1) has been introduced in a newly added section 3 with a question: Why does the minimum static stability in the upper troposphere constrain high ice clouds? The analyses have been added as follows: "Anvil clouds outflow from convection in the upper troposphere. To derive the physical links between convective anvil outflows and static stability, the thermodynamic energy equation at steady state is expressed as (Thompson et al., 2017):

$$\vec{V} \cdot \nabla_h T + \omega S = Q, \text{ (5)}$$

where $\vec{V} \cdot \nabla_h T$ is the horizontal temperature advection, $\omega$ is the vertical velocity in pressure coordinates, $S$ is the stability (i.e., $-\frac{T}{\theta}\frac{\partial \theta}{\partial p}$ in pressure coordinates or $\frac{T}{\theta}\frac{1}{\rho g}\frac{\partial \theta}{\partial z}$ in height coordinates), and $Q$ is the diabatic heating. In the tropics with a weak horizontal gradient of temperature, $\omega$ is forced by diabatic heating divided by stability ($\frac{Q}{S}$) (Mapes and Houze, 1995; Thompson et al., 2017). Furthermore, by combining the continuity equation and Eq. (5), the divergence (D) can be expressed as a function of $Q$ and $S$:

$$D = \frac{\partial \omega}{\partial p} = \frac{\partial}{\partial p}\left(\frac{Q}{S}\right) = \frac{1}{S}\frac{\partial Q}{\partial p} - \frac{Q}{S^2}\frac{\partial S}{\partial p} = \frac{1}{S}\frac{\partial Q}{\partial p}\left(1 - \frac{\partial lnS}{\partial lnQ}\right). \text{ (6)}$$

In the fixed anvil-top temperature (FAT) hypothesis (Hartmann and Larson, 2002), for mass conservation, the level of the maximum divergence in convective regions is constrained by the height of the clear-sky radiative-driven convergence (i.e., the rapid decline of clear-sky radiative cooling that corresponds to the peak value in $\frac{\partial Q}{\partial p}$). Since $\frac{\partial Q}{\partial p}$ in the clear sky is determined by the radiative emission of water vapor and the saturation vapor pressure is largely dependent on temperature, $\frac{\partial Q}{\partial p}$ peaks at constant temperatures in clear-sky regions to limit the detrainment level of convective regions. On the long-term average, convection outflows occur at the level of the fixed temperature of clear-sky regions (Thompson et al., 2017). The FAT hypothesis provides a useful constraint of clear-sky cooling on the level of convective outflow in climate studies. Nevertheless, it is not expected that the level of convective outflow is determined instantaneously by clear-sky radiative cooling, since convection and radiation are not always balanced at short time scales (Tompkins and Craig, 1998). Even at seasonal time scales, Chae and Sherwood (2010) argued that the anvil-top temperature is not perfectly fixed but has a variation of ~5 K, and this variation is related to stability conditions in the upper troposphere. As climate warms, Zelinka and Hartmann (2010) suggested that the increase in stability prevents clouds from rising isothermally and results in a smaller longwave cloud feedback than that predicted by the FAT hypothesis, namely the proportionately higher anvil temperature (PHAT) assumption. These findings are reasonable, since, in Eq. (6), $D$ is a function of both $Q$ and $S$. Specifically, $D$ is

determined by the static stability $S$, the diabatic-heating destabilization $\frac{\partial Q}{\partial p}$, and the changing rate of $S$ due to the variation in diabatic heating $\frac{\partial lnS}{\partial lnQ}$. In the upper troposphere, the capability of the air holding water vapor and latent heat release are limited by low temperatures based on the Clausius-Clapeyron function, and thus $Q$ is largely related to the cloud radiative heating in convective regions (Harrop and Hartmann, 2016; Gasparini et al., 2019; Stubenrauch et al., 2021; Haslehner et al., 2024). On average, the magnitude of the cloud radiative heating rate is 0.42 K/hour (Wall et al., 2020). At short time scales, if $S$ does not respond strongly to the limited variation in $Q$ (but might be more sensitive to the vertical gradient of $Q$) with the term $\frac{\partial lnS}{\partial lnQ} \ll 1$, Eq. (6) can be further simplified as:

$$D \approx \frac{1}{S}\frac{\partial Q}{\partial p}. \text{ (7)}$$

Thus, for the $D$ profiles in convective regions, $D$ is proportional to $\frac{1}{S}$ and the radiative destabilization $\frac{\partial Q}{\partial p}$. Gettelman and Forster (2002) and Folkins (2002) observed that the height of the dθ/dz minimum coincides with the level of the maximum convective outflow. This height of the dθ/dz minimum is also usually used as the dynamical or convective tropopause ~12-14 km or 345 K (Folkins, 2002; Folkins and Martin, 2005; Randel and Jensen, 2013; Babu, 2024), at which convective outflow increases rapidly and temperature profiles starts to increasingly deviates from a moist adiabat (Folkins, 2002). An explanation might be found according to Eq. (7): at the level of the minimum point of $S$, $\frac{1}{S}$ is the largest; thus, as presented in those previous studies, the minimum point of $S$ corresponds to that $D$ increases to approximately the maximum:

$$D_{max} \approx \frac{1}{S_{min}}\frac{\partial Q}{\partial p}. \text{ (8)}$$

The subscripts 'max' and 'min' represent the maximum and minimum points in the vertical direction, respectively. Technically, in consideration of the real $D$ as a function of both $Q$ and $S$, $S_{min}$ and $(\frac{\partial Q}{\partial p})_{max}$ both can contribute to $D_{max}$. If $D$ is strongly controlled by $S$ in convective regions, the level of $S_{min}$ might approximate to the realistic level of $D_{max}$ or correspond to one of peaks in $D$. If this hypothesis is appropriate, according to Eq. (8), the minimum stability in the upper troposphere should be a strong physical constraint on the maximum divergence, and their heights should be approximately coincident in the tropics. In fact, their height consistency has been well confirmed in previous studies (Folkins, 2002; Gettelman and Forster, 2002), but the relationship between the minimum stability and the strength of divergence has been rarely studied. Although $(\frac{\partial Q}{\partial p})_{max}$ could also be very important to contribute to a peak of $D$ and worthy further investigations, the main focus of this work is the effects of $S_{min}$ on $D_{max}$. Here, we define the minimum dθ/dz in the upper troposphere (between 5-18 km) as the estimated anvil-outflow stability (EAS):

$$EAS = (\frac{d\theta}{dz})_{min}. \text{ (9)}$$

Here, dθ/dz is not the only form to express static stability but it has been used most often to connect convective processes with stability regimes (Folkins, 2002; Gettelman and Forster, 2002; Frierson, 2006; Mehta et al., 2008; Sunilkumar et al., 2017; Babu, 2024). Overall, the level of convective outflow is determined by both the FAT and stability, and seems to be predicted by the height of the dθ/dz minimum point (as illustrated in Fig. 1) according to the approximation of Eq. (8) and

observations in many previous studies (Folkins, 2002; Gettelman and Forster, 2002; Randel and Jensen, 2013; Babu, 2024). Moreover, the dθ/dz minimum value, namely the EAS, seems to be more physically relevant to the strength of convective outflow than the UTS. Further observational validation for this hypothesis is laid out in Sect. 4 at a ground-based site and Sect. 5 at a global scale.".

==Thank you again sincerely for providing this framework.==

- ***Use of ground-based radar.*** *The MMCR is not the ideal choice of instrument for detecting cloud top height (CTH). The radar is not very sensitive to small ice crystals, which tend to dominate the upper parts of anvil cirrus. This can make a big difference in retrieved CTH and cloud fraction statistics; see the Key Figures on this page:* [https://climatedataguide.ucar.edu/climate-data/combined-cloudsat-](https://climatedataguide.ucar.edu/climate-data/combined-cloudsat-) [spaceborne-radar-and-calipso-spaceborne-lidar-cloud-fraction-dataset](https://climatedataguide.ucar.edu/climate-data/combined-cloudsat-spaceborne-radar-and-calipso-spaceborne-lidar-cloud-fraction-dataset)*. As a result, the typical CTHs shown in Fig 2 are 1-2 km lower than previous studies using spaceborne lidar, e.g., Berry & Mace (2014; doi: 10.1002/2014JD021458), Hartmann & Berry (2017; doi:10.1002/2017JD026460), Dessler et al (2006; doi:10.1029/2005JD006705), among many others. The same bias can be seen when compared to the DARDAR cloud fraction in Fig 6, where CTHs look close to 15 km.*

  *In addition, the MMCR can become attenuated by precipitation and optically thick clouds (Hollars et al 2004, doi:10.1016/j.atmosres.2004.03.015), which would also bias CTH low in the case of precipitating convection.*

  *This issue does not necessarily invalidate the rest of the analysis, since this is not a study of CTH itself. But it means that UTS, which measures stability between ~13.5-16.5 km, does coincide rather closely with true CTH. This weakens the author's argument that EAS is favorable over UTS because of its closer proximity to the anvil. Nevertheless, the EAS-HCC relationship still seems to be better than UTS-HCC, so the main conclusion is not affected.*

  ==Response:== I agree with the reviewer that the MMCR-detected cloud-top height (CTH) and cloud fractions related to small ice crystals have biases. This bias has been clarified in the revised manuscript. And it has been clarified that the anvil top here refers to the level of main convective outflows but not exactly the anvil top height. Thus, owing to the bias of the MMCR-detected CTH, a concern is whether the main level of the convective outflow is captured by MMCR-detected CTH. Thus, a further validation is presented in Fig. 3 (newly added in the revised manuscript and shown below) on the basis of divergence profiles. The divergence profiles are derived from the EAR5 hourly reanalysis to collocate with the radiosonde observations. In Fig. 3, the divergence strength is inversely proportional to the EAS, and the height of the maximum divergence is close to but below the EAS height. This further supports the EAS constraint on the height and strength of convective outflows.

[Figure]

Figure 3. The composited divergence profiles of ERA5 against the EAS measured by radiosondes at the Manus site. The blue solid line indicates the mean level of the maximum divergence. The blue dashed line indicates the mean height of the dθ/dz minimum.

In the revised manuscript, a discussion has been added as: "some biases might exist in the MMCR-detected ice cloud fraction and height in Fig. 2. Thin cirrus clouds of in situ origin normally form in slow updrafts where the environmental temperature is below -38℃ (Krämer et al., 2016) and account for a large portion of tropical cirrus clouds (Luo and Rossow, 2004). However, these thin cirrus clouds might not be well identified by the MMCR. The MMCR is not sensitive to small ice crystals and is quickly attenuated by precipitation and optically thick clouds (Hollars et al., 2004). This means that some upper parts of thick clouds and thin cirrus clouds could be missed by the MMCR in Fig. 2. Compared with the previous studies (Dessler et al., 2006; Berry and Mace, 2014; Hartmann and Berry, 2017), the ice cloud top height detected by the ground-based MMCR (shown in Fig. 2) is about 1-2 km lower than the cloud top height detected by the spaceborne lidar. Thus, for the relationship between EAS and convective outflows in Eq. (8), further validation is presented in Fig. 3 on the basis of the divergence profiles. The divergence profiles are derived from the EAR5 hourly reanalysis to collocate with the radiosonde observations. Notably, in the reanalysis, the divergence largely relies on the model data to fulfill the consistency with the laws of physics, and thus some bias might exist in the divergence strength. Nevertheless, high precision of the divergence strength is not very necessary for only qualitatively diagnosing the rationality of the EAS constraint on the divergence. In Fig. 3, the divergence strength is inversely proportional to the EAS strength, and the height of the maximum divergence is close to but below the EAS height."

- *Please specify how the moist adiabatic dθ/dz is calculated. Does the author take the observed pressure and temperature at some vertical level and use it calculate the moist adiabatic lapse rate? Or is it found by launching a moist adiabatic parcel profile from some assumed surface conditions? These two methods would give different results. Also, does the calculation use the saturation vapor pressure over liquid throughout the entire troposphere, or is there a transition to ice saturation at cold temperatures?*

**Response:** Moist adiabatic dθ/dz ($\Gamma_m$) can be calculated from the observed temperature and pressure profiles as:

$$\Gamma_m\,(T,p) = \left(\frac{1000}{p}\right)^{\frac{R_a}{c_{pa}}} \cdot \frac{g}{c_{pa}}\left(1 - \frac{1+L_v q_s(T,p)/R_a T}{1+L_v^2 q_s(T,p)/c_{pa}R_v T^2}\right).\,(7)$$

$T$ and $p$ are the radiosonde-detected temperature and pressure. $R_a$ is the specific gas constant of dry air. $R_v$ is the specific gas constant for water vapor. $c_{pa}$ is the specific heat capacity for dry air at constant pressures. $g$ is the gravitational acceleration. $q_s$ is the saturated mass fraction of water vapor. $L_v$ is the latent heat of vaporization.

This has been specified in the revised manuscript. At vertical each level, the observed pressure and temperature were taken in to the Eq. (7) to compute moist adiabatic dθ/dz. The calculation only uses the saturation vapor pressure over liquid throughout the entire troposphere.

**Line Comments**

- *Line 28: I found the wording "Cloud responses to the environmental changes have not been correctly simulated in models" to be a bit odd. We do not know if cloud responses have been correctly simulated, since we do not know the ground truth in future climate scenarios. I'd suggest more precise language for the first sentence of the paper, i.e., something about uncertainty.*

**Response:** Thanks. This sentence has been corrected as: "Cloud responses to environmental changes exhibit uncertainty in models".

- *Line 35 "to the mass" -> "on the mass"*

**Response:** It has been corrected as: "on the mass".

- *Line 36-37: "are the first-order cloud-controlling factor in all scales"….this strikes me as an enormous claim that I have never heard before. All scales, even microphysical? Again, more precise language is needed here.*

**Response:** It has been corrected as: "is a bridge for the interactions between the environment and clouds".

- *Line 44-45 "convection-to-radiation transition" – again, not very precise language. I know what the authors mean here, but many readers may not.*

**Response:** This sentence has been modified as: "a minimum atop the level of convection outflow that results from the tropical tropopause layer (TTL) transition from the convectively dominated troposphere to the radiatively controlled stratosphere"

- *Line 49: "moist adiabat" -> "moist adiabatic ascent"*

**Response:** It has been corrected as: "moist adiabatic ascent".

- *Line 51: "(neutrally buoyant for the air-parcel ascending)" what do the authors mean by this? Is the author trying to say that ascending parcels are neutrally buoyancy? If so, that is not truly the case, just an approximation used in theoretical studies of the lapse rate, e.g. Singh & O'Gorman 2013.*

**Response:** It has been deleted and corrected as: "tropical free-tropospheric dθ/dz is basically determined by the latent heat release".

- *Line 59: "latent heat release and radiation and interactive with the anvil clouds" -> fix grammar*

**Response:** It has been corrected as: "The dθ/dz values are determined by both latent heat release and radiative heating, and the stability interacts with high ice clouds.".

- *Line 104: "with the distance less than 250" – specify that this is the clear-sky distance.*

**Response:** It has been specified.

- *Line 126: Lidar is sensitive to small ice particles in addition to liquid. As written, it sounds like radar detects only ice and lidar detects only liquid. DARDAR is an ice- only product that still relies heavily on the lidar.*

**Response:** This sentence has been rewritten as: "This dataset is based on the combined observations of the Cloud Profiling Radar (CPR) on board the CloudSat satellite and the Cloud-Aerosol Lidar with Orthogonal Polarization instrument (CALIOP) on board the Cloud-Aerosol Lidar and Infrared Pathfinder Satellite Observations (CALIPSO). The CloudSat radar operates at 95 GHz, which is more sensitive to relatively larger cloud particles and can detect the major vertical structure of clouds except very thin clouds such as cirrus. On the contrary, the CALIOP (lidar) operates at the wavelengths of 1024 nm and 532 nm and is more sensitive to relatively smaller particles to detect optically thin clouds, whereas its signal is quickly attenuated for thick clouds. Thus, the CPR and CALIOP combination can provide more accurate full cloud profiles than using either of them individually. When clouds are present, at the level where the temperature is between -40℃ and 0℃, ice particles can be discriminated from supercooled droplets by their relatively higher radar reflectivity but smaller lidar backscatter signals (Hogan et al., 2004; Delanoë and Hogan, 2010). As a result, the radar and lidar combination can benefit the discrimination of cloud phases. In this work, only ice-phase clouds of the DARDAR dataset are considered (see details in Sect. 5.2).".

- *Line 133: shouldn't the vertical resolution be in hPa, not meters?*

**Response:** It has been corrected as: "In vertical direction, the atmospheric profiles in reanalysis have 16 pressure levels between 50 hPa and 500 hPa. The vertical resolution is 50 hPa from 250 to 500 hPa, and is 25 hPa from 50 to 250 hPa.".

- *Line 137: Sentence starting with "The lowest half level…" is unclear. In addition, I think the same*

*symbol is being used to indicate a hyphen as well as a negative sign, which is confusing in this case*

**Response:** This sentence has been rewritten as: "Until both criteria are fulfilled at the half level $j + 1/2$, the exact position of the LRT is linearly interpolated between the levels of $j - 1/2$ and $j + 1/2$:

$$LRT = z_{j-1/2} + \frac{z_{j+1/2} - z_{j-1/2}}{dT/dz_{j+1/2} - dT/dz_{j-1/2}} (-2 - dT/dz_{j-1/2}).$$

Reichler et al. (2003) and Meng et al. (2021) shown that the root-mean-square errors of the reanalysis-based LRT is about 30-40 hPa in extratropics and 10-20 hPa in tropics in comparison to the radiosonde measures. "

In addition, the hyphen and negative signs have been indicated by different symbols: "-" and "−", respectively.

- *Line 144: "existence" -> "significance"*

**Response:** It has been corrected as "significance".

- *Line 146: "The number of independent samples is determined…by the distance between independent samples." Something is off here—you can't use the number of independent samples to determine the number of independent samples. And is this distance referring to spatial or temporal distance? If it is temporal distance, shouldn't the autocorrelation be used to determine the number of independent samples?*

**Response:** Yes, it refers to the temporal distance. It has been revised as: "The number of independent samples is determined based on the e-folding length of autocorrelation".

- *Line 154: caption references a dashed red line but there is no dashed red line*

**Response:** It has been corrected as: "Composites of the environmental dθ/dz (Γ, solid blue and red lines) for cloudy and clear skies, respectively. The dashed blue line represents moist-adiabatic dθ/dz (Γm).".

- *Line 164: absorption of longwave radiation by water vapor.*

**Response:** It has been corrected.

- *Line 175: "is close to the anvil top of the maximum convective outflows"…is the author referencing the anvil top height or the height of max outflow? As they noted earlier, these heights are different.*

**Response:** It refers to the height of max outflow. The EAS is redefined as the estimated anvil-outflow stability. And a specific description has been added as Sect. 3 in the revised manuscript for why it is associated with the level of convective outflows.

- *Fig 3: where are the results of the statistical significance tests for the correlations?*

**Response:** These correlations in Fig. 3 are at the 95% significant level. It has been clarified in the text.

- *Fig 3: Do pabels b, c, e, f show results just for the Manus location, or for the whole 60S-60N study region? It is sometimes hard to follow which instruments are being used in each of the figures (e.g., is ice cloud fraction still from MMCR or now from MODIS). It would be helpful to specify in the caption.*

**Response:** Yes, the results in Fig. 3 are just for the Manus location. The location has been specified in the caption. The use of the instrument has been specified in the colorbar (as shown below). Additionally, a brief introduction and subtitles have been added at the begin of each section to help grasp the goal of each part of analyses.

[Figure]

- *Line 245: "approves" is not the right word here. Perhaps something like "supports the idea that"*

**Response:** It has been modified as: "supports the idea that".

- *Line 254: what does the author mean by "in which only the correlation at the 95% significance level is counted"? Is the daily mean UTS and HCC being computed for the entire HCC regions, and then the correlations are computed using single values for the entire region? Or, are the daily mean correlations being computed for each grid point, and then the R values averaged across all grid points in the region? If it is the latter, I think all R values should be included in the averaging, not just those that meet the significance test.*

**Response:** Daily mean correlations are computed for each grid point, and then the R values are averaged across all grid points in the region. This has been specified in the revised manuscript. It has been corrected and all R values are included in the averaging.

This part has been modified as: "On average, the absolute values of the UTS-HCC correlations over the domains of the HCC larger than 5% and 20% (i.e., the absolute values of correlation are averaged across all grid points in the region) are 0.21 and 0.27, respectively. In contrast, the means of absolute values of the EAS-HCC correlations over the 5% and 20% HCC domains are 0.39 and 0.50, respectively."

- *Line 276: could it be that large UTS forces convection to detrain at lower altitudes, producing more clouds in the 11-13 km range?*

**Response:** Yes, it is possible. I further added the heights of the LRT, EAS and maximum divergence to Figs. 7-8 (the figures are shown in the next page). Larges UTS does correspond to lower altitude of divergence.

- *Line 280: why would a more unstable environment sustain anvil clouds over time?*

**Response:** This sentence has been corrected as: "The relationship between the ice cloud fraction and EAS is physically reasonable and consistent with Eq. (8). An unstable upper-tropospheric environment favors stronger convective outflows to be more likely to produce more high ice clouds.".

- *Line 286: is this relationship indeed linear? This is hard to tell from the color scale used in Figs 6-7.*

**Response:** To better show the relationship, the contours of ice cloud fraction have been added in Figs. 7-8 in the revised manuscript (Figs. 6-7 in the previous edition), as shown below.

[Figure]

Figure 7. Relationship between the DARDAR-detected ice cloud fraction profiles and the ERA5-based UTS and EAS over oceans. (a-c) The composited ice cloud fraction in each bin of the UTS over tropics, subtropics and midlatitude, respectively. (d-f) The composited ice cloud fraction in each bin of the EAS over tropics, subtropics and midlatitude, respectively. The blue solid and dashed lines indicate the height of the maximum divergence and the height of the EAS, respectively. The green dashed lines are the height of the LRT. The grey contours refer to the ice cloud fraction.

[Figure]

Figure 8. Similar to Fig. 7 but over the land.

- *Line 293: "approved" -> showed*

**Response:** It has been corrected.

- *Line 298: "approves" -> shows*

**Response:** It has been corrected.

- *Line 306: servers -> serves*

**Response:** it has been corrected.

**Reference**

Babu, S. R.: Convective tropopause over the tropics: Climatology, seasonality, and inter-annual variability inferred from long-term FORMOSAT-3/COSMIC-1 RO data, Atmospheric Research, 298, 107159, https://doi.org/10.1016/j.atmosres.2023.107159, 2024.

Berry, E. and Mace, G. G.: Cloud properties and radiative effects of the Asian summer monsoon derived from A‑Train data, Journal of Geophysical Research: Atmospheres, 119, 9492-9508, 10.1002/2014jd021458, 2014.

Bony, S., Stevens, B., Coppin, D., Becker, T., Reed, K. A., Voigt, A., and Medeiros, B.: Thermodynamic control of anvil cloud amount, Proc Natl Acad Sci U S A, 113, 8927-8932, 10.1073/pnas.1601472113, 2016.

Chae, J. H. and Sherwood, S. C.: Insights into Cloud-Top Height and Dynamics from the Seasonal Cycle of Cloud-Top Heights Observed by MISR in the West Pacific Region, Journal of the Atmospheric Sciences, 67, 248-261, 10.1175/2009jas3099.1, 2010.

Delanoë, J. and Hogan, R. J.: Combined CloudSat‑CALIPSO‑MODIS retrievals of the properties of ice clouds, Journal of Geophysical Research: Atmospheres, 115, 10.1029/2009jd012346, 2010.

Dessler, A. E., Palm, S. P., and Spinhirne, J. D.: Tropical cloud‑top height distributions revealed by the Ice, Cloud, and Land Elevation Satellite (ICESat)/Geoscience Laser Altimeter System (GLAS), Journal of Geophysical Research: Atmospheres, 111, 10.1029/2005jd006705, 2006.

Folkins, I.: Origin of Lapse Rate Changes in the Upper Tropical Troposphere, Journal of the Atmospheric Sciences, 59, 992-1005, 10.1175/1520-0469(2002)059<0992:Oolrci>2.0.Co;2, 2002.

Folkins, I. and Martin, R. V.: The Vertical Structure of Tropical Convection and Its Impact on the Budgets of Water Vapor and Ozone, Journal of the Atmospheric Sciences, 62, 1560-1573, 10.1175/jas3407.1, 2005.

Frierson, D. M. W.: Robust increases in midlatitude static stability in simulations of global warming, Geophysical Research Letters, 33, 10.1029/2006gl027504, 2006.

Gasparini, B., Blossey, P. N., Hartmann, D. L., Lin, G., and Fan, J.: What Drives the Life Cycle of Tropical Anvil Clouds?, Journal of Advances in Modeling Earth Systems, 11, 2586-2605, 10.1029/2019ms001736, 2019.

Gasparini, B., Sokol, A. B., Wall, C. J., Hartmann, D. L., and Blossey, P. N.: Diurnal Differences in Tropical Maritime Anvil Cloud Evolution, Journal of Climate, 35, 1655-1677, 10.1175/jcli-d-21-0211.1, 2022.

Gettelman, A. and Forster, P. M. d. F.: A Climatology of the Tropical Tropopause Layer, Journal of the Meteorological Society of Japan. Ser. II, 80, 911-924, 10.2151/jmsj.80.911, 2002.

Harrop, B. E. and Hartmann, D. L.: The role of cloud radiative heating within the atmosphere on the high cloud amount and top-of-atmosphere cloud radiative effect, Journal of Advances in Modeling Earth Systems, 8, 1391-1410, https://doi.org/10.1002/2016MS000670, 2016.

Hartmann, D. L. and Berry, S. E.: The balanced radiative effect of tropical anvil clouds, Journal of Geophysical Research: Atmospheres, 122, 5003-5020, 10.1002/2017jd026460, 2017.

Hartmann, D. L. and Larson, K.: An important constraint on tropical cloud‑climate feedback, Geophysical Research Letters, 29, 10.1029/2002gl015835, 2002.

Haslehner, K., Gasparini, B., and Voigt, A.: Radiative Heating of High-Level Clouds and Its Impacts on Climate, Journal of Geophysical Research: Atmospheres, 129, e2024JD040850, https://doi.org/10.1029/2024JD040850, 2024.

Highwood, E. J. and Hoskins, B. J.: The tropical tropopause, Quarterly Journal of the Royal Meteorological Society, 124, 1579-1604, 10.1002/qj.49712454911, 1998.

Hogan, R. J., Behera, M. D., O'Connor, E. J., and Illingworth, A. J.: Estimate of the global distribution of stratiform supercooled liquid water clouds using the LITE lidar, Geophysical Research Letters, 31, Artn L05106
10.1029/2003gl018977, 2004.

Hollars, S., Fu, Q., Comstock, J., and Ackerman, T.: Comparison of cloud-top height retrievals from ground-based 35 GHz MMCR and GMS-5 satellite observations at ARM TWP Manus site, Atmospheric Research, 72, 169-186, 10.1016/j.atmosres.2004.03.015, 2004.

Hong, Y. and Di Girolamo, L.: Cloud phase characteristics over Southeast Asia from A-Train

satellite observations, Atmos. Chem. Phys., 20, 8267-8291, 10.5194/acp-20-8267-2020, 2020.

Krämer, M., Rolf, C., Luebke, A., Afchine, A., Spelten, N., Costa, A., Meyer, J., Zöger, M., Smith, J., Herman, R. L., Buchholz, B., Ebert, V., Baumgardner, D., Borrmann, S., Klingebiel, M., and Avallone, L.: A microphysics guide to cirrus clouds – Part 1: Cirrus types, Atmos. Chem. Phys., 16, 3463-3483, 10.5194/acp-16-3463-2016, 2016.

Li, Y., Thompson, D. W. J., Stephens, G. L., and Bony, S.: A global survey of the instantaneous linkages between cloud vertical structure and large‐scale climate, Journal of Geophysical Research: Atmospheres, 119, 3770-3792, 10.1002/2013jd020669, 2014.

Lilly, D. K.: Cirrus Outflow Dynamics, Journal of the Atmospheric Sciences, 45, 1594-1605, 10.1175/1520-0469(1988)045<1594:Cod>2.0.Co;2, 1988.

Luo, Z. and Rossow, W. B.: Characterizing Tropical Cirrus Life Cycle, Evolution, and Interaction with Upper-Tropospheric Water Vapor Using Lagrangian Trajectory Analysis of Satellite Observations, Journal of Climate, 17, 4541-4563, 10.1175/3222.1, 2004.

Maleska, S., Smith, K. L., and Virgin, J.: Impacts of Stratospheric Ozone Extremes on Arctic High Cloud, Journal of Climate, 33, 8869-8884, 10.1175/jcli-d-19-0867.1, 2020.

Mapes, B. E. and Houze, R. A.: Diabatic Divergence Profiles in Western Pacific Mesoscale Convective Systems, Journal of the Atmospheric Sciences, 52, 1807-1828, 10.1175/1520-0469(1995)052<1807:Ddpiwp>2.0.Co;2, 1995.

Mehta, S. K., Murthy, B. V. K., Rao, D. N., Ratnam, M. V., Parameswaran, K., Rajeev, K., Raju, C. S., and Rao, K. G.: Identification of tropical convective tropopause and its association with cold point tropopause, Journal of Geophysical Research: Atmospheres, 113, 10.1029/2007jd009625, 2008.

Meng, L., Liu, J., Tarasick, D. W., and Li, Y.: Biases of Global Tropopause Altitude Products in Reanalyses and Implications for Estimates of Tropospheric Column Ozone, Atmosphere, 12, 10.3390/atmos12040417, 2021.

Munchak, L. A. and Pan, L. L.: Separation of the lapse rate and the cold point tropopauses in the tropics and the resulting impact on cloud top-tropopause relationships, Journal of Geophysical Research: Atmospheres, 119, 7963-7978, 10.1002/2013jd021189, 2014.

Randel, W. J. and Jensen, E. J.: Physical processes in the tropical tropopause layer and their roles in a changing climate, Nature Geoscience, 6, 169-176, 10.1038/ngeo1733, 2013.

Randel, W. J., Seidel, D. J., and Pan, L. L.: Observational characteristics of double tropopauses, J Geophys Res-Atmos, 112, Artn D07309
10.1029/2006jd007904, 2007.

Reichler, T., Dameris, M., and Sausen, R.: Determining the tropopause height from gridded data, Geophysical Research Letters, 30, 10.1029/2003gl018240, 2003.

Schmidt, T., Beyerle, G., Heise, S., Wickert, J., and Rothacher, M.: A climatology of multiple tropopauses derived from GPS radio occultations with CHAMP and SAC‐C, Geophysical Research Letters, 33, 10.1029/2005gl024600, 2006.

Seidel, D. J., Ross, R. J., Angell, J. K., and Reid, G. C.: Climatological characteristics of the tropical tropopause as revealed by radiosondes, Journal of Geophysical Research: Atmospheres, 106, 7857-7878, 10.1029/2000jd900837, 2001.

Stubenrauch, C. J., Caria, G., Protopapadaki, S. E., and Hemmer, F.: 3D radiative heating of tropical upper tropospheric cloud systems derived from synergistic A-Train observations and machine learning, Atmos. Chem. Phys., 21, 1015-1034, 10.5194/acp-21-1015-2021, 2021.

Sunilkumar, S. V., Muhsin, M., Venkat Ratnam, M., Parameswaran, K., Krishna Murthy, B. V., and Emmanuel, M.: Boundaries of tropical tropopause layer (TTL): A new perspective based on thermal and stability profiles, Journal of Geophysical Research: Atmospheres, 122, 741-754, 10.1002/2016jd025217, 2017.

Thompson, D. W. J., Bony, S., and Li, Y.: Thermodynamic constraint on the depth of the global tropospheric circulation, Proc Natl Acad Sci U S A, 114, 8181-8186, 10.1073/pnas.1620493114, 2017.

Tinney, E. N., Homeyer, C. R., Elizalde, L., Hurst, D. F., Thompson, A. M., Stauffer, R. M., Vömel, H., and Selkirk, H. B.: A Modern Approach to a Stability-Based Definition of the Tropopause, Monthly Weather Review, 150, 3151-3174, 10.1175/mwr-d-22-0174.1, 2022.

Tompkins, A. M. and Craig, G. C.: Time‐scales of adjustment to radiative‐convective equilibrium

in the tropical atmosphere, Quarterly Journal of the Royal Meteorological Society, 124, 2693-2713, 10.1002/qj.49712455208, 1998.

Wall, C. J., Norris, J. R., Gasparini, B., Smith, W. L., Thieman, M. M., and Sourdeval, O.: Observational Evidence that Radiative Heating Modifies the Life Cycle of Tropical Anvil Clouds, Journal of Climate, 33, 8621-8640, 10.1175/jcli-d-20-0204.1, 2020.

Wilson Kemsley, S., Ceppi, P., Andersen, H., Cermak, J., Stier, P., and Nowack, P.: A systematic evaluation of high-cloud controlling factors, Atmos. Chem. Phys., 24, 8295-8316, 10.5194/acp-24-8295-2024, 2024.

Yuan, J., Houze, R. A., and Heymsfield, A. J.: Vertical Structures of Anvil Clouds of Tropical Mesoscale Convective Systems Observed by CloudSat, Journal of the Atmospheric Sciences, 68, 1653-1674, 10.1175/2011jas3687.1, 2011.

Zelinka, M. D. and Hartmann, D. L.: Why is longwave cloud feedback positive?, Journal of Geophysical Research, 115, 10.1029/2010jd013817, 2010.

---

## Author Comment (AC2)

**Response to Anonymous Referee #2**

*Referee #2: In this manuscript, the author tried to explore the relationship between high-level anvil clouds and static stability through a novel approach known as estimated anvil-top stability (EAS), which is based on the minimum value of dθ/dz. The author claims that the upper-tropospheric stability (UTS) method, which relies on the lapse rate tropopause, underestimates the effect of stability on the anvil, while EAS provides a more accurate relationship. The findings further indicate that EAS has a stronger correlation with anvil clouds than UTS. This proposed method may be useful in understanding the factors that control cloud structure and composition. The topic of the research is interesting and the analysis results would be worth a concise publication. Though the manuscript is scientifically sound enough, the presentation style needs to be improved. Overall, the manuscript requires major revisions. I had the chance to read the comments of Anonymous Reviewer #2 and I do share all his/her general comments.*

Response: We thank anonymous referee for reviewing our manuscript and very helpful comments to modify the manuscript. We have responded to all comments, and carefully improved the presentation of the manuscript accordingly.

***Comments:***

1. *What I can see as one of the major problems of the manuscript is that it lacks clarity in many places in its current form. Several sentences are not clear, please revise.*

Response: The revised manuscript has been better reorganized with a brief introduction at the beginning of each section to help grasp the goal of analyses and to improve the clarity. More descriptions have been added to clarify the results.

The manuscript has been better clarified and specified according to the following comments.

Professional editing service has been pursued to guarantee an appropriate language use.

2. *Is it 3 km moving smoothing? What is the basis for selecting 3-km smoothing? What is the final vertical resolution of radiosonde data?*

Response: Yes, it is 3-km moving smoothing. The selection of the smoothing window is empirical. The smoothing effects have been tested in König et al. (2019), which suggested that 3-km smoothing is well-behaved. The final vertical resolution is still 10 meters. This has been clarified in the revised manuscript.

3. *Is the LRT derived from this 3 km smoothed temperature profile as well? What is the reliability of the relationship established by the results obtained?*

Response: Yes, the LRT is derived from the 3-km smoothed temperature profiles. As tested in König et al. (2019), 3-km smoothing will result in a bias no more than 500 m for the LRT. This has been clarified in the revised manuscript.

4. *The author may clearly describe how the Anvil cloud is categorized in this manuscript. What are the limitations of the MMCR for detecting the anvil clouds?*

**Response:** The MMCR is not sensitive to small ice crystals and is quickly attenuated by precipitation and optically thick clouds (Hollars et al., 2004). It means that the upper parts of thick clouds and thin cirrus clouds could be missed by the MMCR in Fig. 2. In comparison to the previous studies (Dessler et al., 2006; Berry and Mace, 2014; Hartmann and Berry, 2017), the ice cloud top height detected by the ground-based MMCR (shown in Fig. 2) is about 1-2 km lower than the cloud top height detected by the spaceborne lidar.

Nevertheless, the anvil top in this work just refers to the level of main convective outflows but not exactly the anvil top height. Thus, this missing upper parts of thick clouds and thin cirrus clouds may not influence our analyses, but do need further validation.

Thus, for the relationship between EAS and convective outflows, a further validation is presented in Fig. 3 (as shown below) on the basis of divergence profiles. The divergence profiles are derived from the EAR5 hourly reanalysis to collocate with the radiosonde observations. In Fig. 3, the divergence strength is inversely proportional to the EAS, and the height of the maximum divergence is close to but below the EAS height. This further supports the EAS constraint on the height and strength of convective outflows.

[Figure]

Figure 3. The composited divergence profiles of ERA5 against the EAS measured by radiosondes at the Manus site. The blue solid line indicates the mean level of the maximum divergence. The blue dashed line indicates the mean height of the dθ/dz minimum.

5. *What is the time frame for the ERA-5 data and other satellite measurements? Does it align with the radiosonde data from 2001 to 2011? The author should provide clarification on this matter.*

**Response:** In Sect. 4 of the revised manuscript (the Sect. 3 in the previous edition), only the Manus ground-based site during 2001-2011 is investigated. The hourly EAR5 data during the same period at the grid point of 147.5°E and 2.0°S are used. At this site, the MMCR is used to detect clouds and none of other satellite is used. This has been clarified in the main text.

In Sect. 5 of the revised manuscript, the satellite measurements and ERA5 data both in 2007 are used for investigating the relationship between global high clouds and stability. The EAR5 data has 1-hour and 0.5° resolutions. This has been clarified in the main text.

6. *Similarly, the author needs to provide the spatial gridding of each data set in the study. What is the spatial resolution for ERA-5, CERES, and DARDAR? While using multiple data sets of observation, reanalysis, and satellite data, it is suggested that it should be gridded to a uniform resolution for better comparison.*

**Response:** The CERES data has 1-hour and 1° resolution, centered at 0.5°, 1.5°, …. The ERA5 has 1-hour and 0.5° resolution. DARDAR provides instantaneous cloud profiles. For matching the EAR5 and CERES dataset, the ERA5 profiles are averaged to 1° resolution, centered at 0.5°, 1.5°, …. To collocate the EAR5 of 0.5° resolution and DARDAR datasets, the instantaneous DARDAR cloud profiles within 0.25° and half an hour of each ERA5 grid points are used to represent the cloud condition of the ERA5 grid point. This has been specified in the revised manuscript.

7. *What was the horizontal drift of the balloon while comparing the cloud fraction of MMCR?*

**Response:** Owing to the balloon drift, the mean horizontal distance between the balloon location and the MMCR at the cloud top height is 13.1 km.

8. *How are the lower-level thin cirrus ice crystals accounted for if the ice clouds are identified based on cloud top temperature?*

**Response:** As suggested in Krämer et al. (2016), thin in situ origin cirrus clouds are normally formed below -38℃ with slow updraft. But those thin cirrus clouds are not well identified by the MMCR, since the MMCR is not sensitive to small ice crystals. This uncertainty has been clarified in the main text of the revised manuscript.

Due to the limitation of the MMCR on detecting those thin cirrus clouds, a further validation has been added for the relationship between the EAS and ERA5-based divergence (please see the response to the comment #4). The focus of this paper is further clarified as: the EAS is a constraint on the strength and height of convective outflows, and thereby constrains high ice clouds related to convection.

9. *The methodology to estimate the moist adiabatic from observation and model datasets used in this study may be explained.*

**Response:** Moist adiabatic dθ/dz ($\Gamma_m$) is calculated from the radiosonde-observed temperature and pressure profiles as:

$$\Gamma_m\,(T,p) = (\frac{1000}{p})^{\frac{R_a}{c_{pa}}} \cdot \frac{g}{c_{pa}}\,(1 - \frac{1+L_v q_S(T,p)/R_a T}{1+L_v^2 q_S(T,p)/c_{pa}R_v T^2}).\ (7)$$

$T$ and $p$ are the radiosonde-detected temperature and pressure. $R_a$ is the specific gas constant of dry air. $R_v$ is the specific gas constant for water vapor. $c_{pa}$ is the specific heat capacity for dry air at constant pressures. $g$ is the gravitational acceleration. $q_s$ is the saturated mass fraction of water vapor. $L_v$ is the latent heat of vaporization.

This has been specified in the revised manuscript. In Fig. 2a, at each vertical level, the observed pressure and temperature were taken in to the Eq. (7) to compute moist adiabatic dθ/dz.

10. *The height of the minimum potential temperature gradient (**Fig. 4b**), commonly known as the convective outflow level or convective tropopause, has been extensively studied and documented by numerous researchers and needs to be included and discussed in the present study (see the reference).*

Response: Thank you for providing these references. They have been added and discussed in the main text: The EAS is the stability at the dynamical tropopause, which is the lower boundary of the TTL and corresponds to the height of convective outflows (Sunilkumar et al., 2013; Babu, 2024; Randel and Jensen, 2013); In Fig. 5b, the height of the EAS is about 10-13 km in the deep tropics, and is consistent with the radiosonde and GPS RO observations (Gettelman and Forster, 2002; Sunilkumar et al., 2017; Biondi et al., 2012; Sunilkumar et al., 2013; Xian and Fu, 2015; Babu, 2024).

*Technical corrections:*

*The English language used in the manuscript needs to be checked by professionals who are native English speakers.*

**Response:** Professional editing service has been used to improve the language use.

*Line 28: "Cloud responses to the environmental changes have not been correctly simulated in models" may be rewritten avoiding concluding statements.*

**Response:** It has been rewritten as: "Cloud responses to environmental changes have uncertainty in models".

*Line 197: 'Nevertheless, the high-resolution radiosondes are limited at islands and coastal sites or during short-term field campaigns.' What about using the available high-resolution GNSS RO data? see the attached references.*

**Response:** It has been revised as: "Observations from the Global Navigation Satellite System – Radio Occultation (GNSS-RO) and reanalysis are both available to pursue the general global and climate analyses. GNSS-RO observations can provide temperature profiles with the vertical resolution of 100 m for investigating the tropospheric and stratospheric thermal stratifications (Biondi et al., 2012; Sunilkumar et al., 2013; Xian and Fu, 2015; Ho et al., 2020; Babu, 2024). The reanalysis can provide atmospheric data with hourly resolution and covers a full period from 1940 to present (Hersbach et al., 2020), although the vertical resolution of the reanalysis is coarse."

*Line 225: How is the ERA-5 data on pressure levels used to identify the height of LRT? Figure 1 The tick labels are missing on the y-axis.*

**Response:** The ERA5 temperature, geopotential and divergence profiles are used in this work. $\theta$ is computed via temperature and pressure. The ERA5 $dT/dz$ and $d\theta/dz$ profiles are computed from the ERA5 temperature and geopotential profiles at the half levels:

$$\frac{dT}{dz}_{i+1/2} = \frac{T_{i+1}-T_i}{z_{i+1}-z_i}, (1)$$

$$\frac{d\theta}{dz}_{i+1/2} = \frac{\theta_{i+1}-\theta_i}{z_{i+1}-z_i}, (2)$$

$$z_{i+1/2} = \frac{z_{i+1}+z_i}{2}, (3)$$

where $T$, $z$ and $\theta$ are the ERA5 temperature, geopotential height and potential temperature, respectively. The subscripts '$i+1$' and '$i$' represent two adjacent levels in the ERA5 atmospheric profiles. The subscript '$i+1/2$' represents the gradient at the half level. The method of calculating the WMO LRT is consistent with that proposed in Reichler et al. (2003):

i. Linearly interpolate the profiles of $(dT/dz)_{i+1/2}$ and $(d\theta/dz)_{i+1/2}$ to obtain the continuous profiles of $dT/dz$ and $d\theta/dz$ at the 100-m resolution;

ii. Search for the lowest half level of $-(dT/dz)_{i+1/2}$ less than 2 K/km above 5 km;

iii. Compute the mean of $-dT/dz$ for a 2-km deep layer above the half level that is located in the second step, and if it is greater than 2 K/km, repeat the second step to search upward further for the half level whose $-(dT/dz)_{i+1/2}$ is less than 2 K/km, until both of the criteria are fulfilled at the half level $j + 1/2$;

iv. Compute the exact position of the LRT between the levels of $j - 1/2$ and $j + 1/2$ via linear interpolation:

$$LRT = z_{j-1/2} + \frac{z_{j+1/2} - z_{j-1/2}}{dT/dz_{j+1/2} - dT/dz_{j-1/2}}(-2 - dT/dz_{j-1/2}). \quad (4)$$

Reichler et al. (2003) and Meng et al. (2021) reported that the root-mean-square errors of the reanalysis-based LRT are 30-40 hPa in the extratropics and 10-20 hPa in the tropics in comparison with radiosonde measurements.

This method of the LRT identification has been clarified in the main text.

The tick labels have been added in Fig. 1 as shown below.

[Figure]

**References:**

Biondi, R., W. J. Randel, S.-P. Ho, T. Neubert, and S. Syndergaard, 2012: Thermal structure of intense convective clouds derived from GPS radio occultations. Atmos. Chem. Phys., 12, 5309– 5318, https://doi.org/10.5194/acp-12-5309-2012.

Sunilkumar, S.V., Babu, A., Parameswaran, K., 2013. Mean structure of the tropical tropopause and its variability over the Indian longitude sector. Clim. Dyn. 40, 1125–1140. https://doi.org/10.1007/s00382-012-1496-8.

Ravindra Babu, S. "Convective Tropopause Over the Tropics: Climatology, Seasonality, and Inter- Annual Variability Inferred from Long-Term FORMOSAT-3/COSMIC-1 RO Data," Atmospheric Research, vol. 298, https://doi.org/10.1016/j.atmosres.2023.107159, 2024.

Ho, S.-P., and Coauthors, 2020: The COSMIC/FORMOSAT-3 radio occultation mission after 12 years: Accomplishments, remaining challenges, and potential impacts of COSMIC-2. Bull. Amer. Meteor. Soc., 101, E1107–E1136, https://doi.org/10.1175/BAMS-D-18-0290.1.

Xian, T., and Y. Fu, 2015: Characteristics of tropopause-penetrating convection determined by TRMM

*and COSMIC GPS radio occultation measurements. J. Geophys. Res. Atmos., 120, 7006– 7024,* [https://doi.org/10.1002/2014JD022633](https://doi.org/10.1002/2014JD022633).